# Self-activating anti-infection implant

Jieni Fu[1,6], Weidong Zhu[2,3,6], Xiangmei Liu [2✉], Chunyong Liang[2✉], Yufeng Zheng [4], Zhaoyang Li[1],
Yanqin Liang[1], Dong Zheng[5], Shengli Zhu [1], Zhenduo Cui[1] & Shuilin Wu [1,3✉]

Clinically, it is difficult to endow implants with excellent osteogenic ability and antibacterial activity simultaneously. Herein, the self-activating implants modified with hydroxyapatite (HA)/MoS$_2$ coating are designed to prevent *Staphylococcus aureus* (*S. aureus*) and *Escherichia coli* (*E. coli*) infections and accelerate bone regeneration simultaneously. The electron transfer between bacteria and HA/MoS$_2$ is triggered when bacteria contacted with the material. RNA sequencing data reveals that the expression level of anaerobic respiration–related genes is up-regulated and the expression level of aerobic respiration–related genes is down-regulated when bacteria adhere to the implants. HA/MoS$_2$ presents a highly effective antibacterial efficacy against both *S. aureus* and *E. coli* because of bacterial respiration–activated metabolic pathway changes. Meanwhile, this coating promotes the osteoblastic differentiation of mesenchymal stem cells by altering the potentials of cell membrane and mitochondrial membrane. The proposed strategy exhibits great potential to endow implants with self-activating anti-infection performance and osteogenic ability simultaneously.

[1] School of Materials Science & Engineering, the Key Laboratory of Advanced Ceramics and Machining Technology by the Ministry of Education of China, Tianjin University, 300072 Tianjin, China. [2] School of Life Science and Health Engineering, Hebei University of Technology, Xiping Avenue 5340, Beichen District, 300401 Tianjin, China. [3] Biomedical Materials Engineering Research Center, Collaborative Innovation Center for Advanced Organic Chemical Materials Co-constructed by the Province and Ministry, Hubei Key Laboratory of Polymer Materials, Ministry-of-Education Key Laboratory for the Green Preparation and Application of Functional Materials, School of Materials Science and Engineering, Hubei University, 430062 Wuhan, China. [4] School of Materials Science & Engineering, State Key Laboratory for Turbulence and Complex System, Peking University, 100871 Beijing, China. [5] Department of Orthopaedics, Union Hospital, Tongji Medical College, Huazhong University of Science and Technology, 430022 Wuhan, China. [6] These authors contributed equally: Jieni Fu, Weidong Zhu. ✉email: liuxiangmei1978@163.com; liangchunyong@hebut.edu.cn; shuilinwu@tju.edu.cn

Bacterial infections and insufficient osseointegration with the host tissue delay patient recovery time and increase postoperative morbidity[1]. The key to success of implantation in the early stage is the rapid osteoblastic differentiation of osteoblasts on the implants and the prevention of bacterial infections simultaneously. Thus, many strategies are used to address this issue, for example, integrating antibacterial agents such as organic bactericides or antibiotics[2], inorganic antibacterial agents like silver/zinc[3], silver/strontium[4], or copper/magnesium[5], with osteogenic peptide or growth factors onto the surface of implants. However, these strategies can induce tissue toxicity to some extent or bacterial resistance due to the sustained release of antibacterial agents, which limits their applications in the clinical field. Therefore, it is essential to develop safe strategies without using antibiotics to prevent bacterial infections and simultaneously enhance osseointegration.

Electron transfer is crucial for the energy metabolism of living cells. Disturbing the process can stimulate the generation of intracellular reactive oxygen species (ROS)[6]. During bacterial respiration, bacteria can transfer endogenous electrons to extracellular electron acceptors via outer membrane $c$-type cytochromes, conductive bacterial nanowires, and/or self-secreted flavins[7,8]. During cellular respiration, cells can generate energy in the form of adenosine triphosphate (ATP) through electron transfer within the inner membrane of mitochondria[9]. This structure difference between cells and bacteria in terms of transferring electrons during metabolism provides a platform to modulate cell fate and bacterial metabolism. This kind of platform depending upon metabolism differences can induce the osteogenic differentiation of mesenchymal stem cells (MSCs) and kill pathogenic bacteria by changing intact metabolic pathways. Hydroxyapatite (HA) is broadly applied in the osseointegration of dental and orthopedic implants because of its osteoconductivity[10]. The bacteria adhered to the surface of inert implants tend to form a compact biofilm and lead to implant failure[11]. It has been reported that besides its excellent biocompatibility, molybdenum sulfide (MoS$_2$) can exhibit metallic or semiconducting properties by tuning its structure, which makes it be used in biomedical filed including anti-cancer therapy, antibacterial therapy, and diagnostics[12–16]. Additionally, the unique physical and photo-electrical properties can endow this material with excellent bactericidal ability against both Gram-positive and Gram-negative bacteria[13,17,18]. Furthermore, 2D MoS$_2$ can have a tunable electronic energy state due to its tunable nanosheet-like structure[19]. Also, nanostructured MoS$_2$ can benefit the separation of electron-hole pairs by not only decreasing the distances of electrons and holes to move to the surface of the materials but also increasing the active sites[17]. It is well known that Mo is the basic trace element of many enzymes, and S is a common biological element in cells, which make MoS$_2$ potentially attractive for biomedical applications[20]. Meanwhile, the conductive circuit might be formed between HA/MoS$_2$ coating and the bacteria. However, little is known about the biological role of HA/MoS$_2$ coating in the process of electron transfer between bacteria and the material.

In this study, we hypothesized that the HA/MoS$_2$ coating could endow the metallic implants with excellent osteogenic ability and antibacterial activity simultaneously. The Ti6Al4V (Ti6) implant modified with HA/MoS$_2$ coating was prepared by laser cladding and chemical vapor deposition (CVD). In addition, the HA/MoS$_2$ coating could effectively eradicate bacteria by extracting electrons from bacteria and simultaneously accelerate bone regeneration by promoting the osteogenic differentiation of MSCs (Fig. 1). When bacteria adhered to HA/MoS$_2$ coating on Ti6, the potential difference between HA/MoS$_2$-Ti6 and bacteria made the electrons from bacterial membranes transfer to the surface of HA/MoS$_2$ coating. The process took advantage of the energy from metabolic process of *S. aureus* and *E. coli* without extra energy to trigger antibacterial activity of HA/MoS$_2$-Ti6. It was a self-activating

process of anti-infection. Meanwhile, the HA/MoS$_2$-Ti6 could enhance the viability of MSCs and promote the osteogenic differentiation of MSCs by upregulating Ca$^{2+}$ level. The discovery of self-activating anti-infection implants is expected to spur the development of orthopedic implants in clinics.

## Results and discussion

### Characterization and antibacterial performance of HA/MoS$_2$–Ti6.
The HA/MoS$_2$–Ti6 was prepared via laser cladding and CVD. Briefly, the MoS$_2$ and HA powders were introduced to the surface of Ti6 implants via the laser cladding. Then, the sulfur powders were sublimated at a high temperature (750 °C) and the sublimed sulfur reacted with the modified implants by laser cladding process to form HA/MoS$_2$–Ti6 (Fig. 2a). The scanning electron microscopy (SEM) image was obtained to demonstrate that Ti6 substrate was completely covered by HA/MoS$_2$ coating (Fig. 2b). The elemental mapping images of HA/MoS$_2$ were shown in Supplementary Fig. 1, which displayed a homogeneous distribution of Ca, P, Mo, and S elements in the coating. The cross-sectional image of HA/MoS$_2$–Ti6 (Fig. 2c) showed that a cladding layer formed on the surface (marked by the yellow broken-line area), indicating that the coating was successfully cladded on the Ti6 substrate. Meanwhile, the Raman spectrum showed two salient peaks, corresponding to the in-plane E$^1_{2g}$ (382.2 cm$^{-1}$) and out-of-plane A$_{1g}$ (404.5 cm$^{-1}$) vibrations of MoS$_2$, indicating the successful sulfuration (Supplementary Fig. 2)[21]. To further analyze the surface chemical compositions, the samples were characterized by X-ray photoelectron spectroscopy (XPS). As shown in Fig. 2d, there were obvious peaks of Ca and P in the HA–Ti6 spectrum and peaks of Mo and S in the MoS$_2$–Ti6 spectrum. Meanwhile, the peaks of Ca, P, Mo, and S were clearly shown in the HA/MoS$_2$–Ti6 spectrum. Supplementary Fig. 3 showed the Mo 3d and S 2p spectra obtained from HA/MoS$_2$–Ti6 spectrum. The peaks at 229.2 eV and 232.4 eV corresponded to Mo 3d$_{5/2}$ and Mo 3d$_{3/2}$ orbitals severally, whereas the peaks at 162.0 eV and 163.2 eV corresponded to S 2p$_{3/2}$ and S 2p$_{1/2}$ orbitals, which belonged to MoS$_2$[22]. The result demonstrated the existence of MoS$_2$ in the coating. X-ray diffractometry (XRD) was further performed to analyze the crystalline structure. The planes of the (002) and (110) were the representative peaks of MoS$_2$[23]. And the planes of the (002) and (211) were assigned to HA[24]. These results revealed that HA was successfully cladded with MoS$_2$ (Fig. 2e). The electron paramagnetic resonance (EPR) measurement was further performed to characterize the vacancy in HA/MoS$_2$–Ti6 (Supplementary Fig. 4). There were no signals of vacancies for HA/MoS$_2$–Ti6, suggesting that the vacancies were not formed after the treatment of laser cladding and CVD. Sulfur powders would be sublimated under high temperatures, and the sulfur powder microenvironment limited the formation of vacancies. The water contact angle was measured to evaluate the surface hydrophilicity or hydrophobicity. Supplementary Fig. 5 showed that the contact angle of Ti6 was 81.6°. After the treatment of laser cladding, the contact angle of HA–Ti6, MoS$_2$-Ti6, and HA/MoS$_2$-Ti6 was reduced to 33.7°, 34.6°, and 29.7°, respectively. The improvement of the surface hydrophilicity of HA–Ti6, MoS$_2$-Ti6, and HA/MoS$_2$-Ti6 was ascribed to the enhanced surface roughness of implants caused by laser cladding process. Meanwhile, the hydroxyl groups in HA could further enhance surface hydrophilicity[25]. It was believed that the hydrophilicity of scaffolds facilitated cell adhesion by increasing the affinity between the proteins on cell membrane and the matrix[26]. The transmission electron microscope (TEM) image obtained from the scraped powders of the HA/MoS$_2$–Ti6 was shown in Fig. 2f, which exhibited an irregular shape. The high-resolution TEM image from HA/MoS$_2$–Ti6 was further obtained,

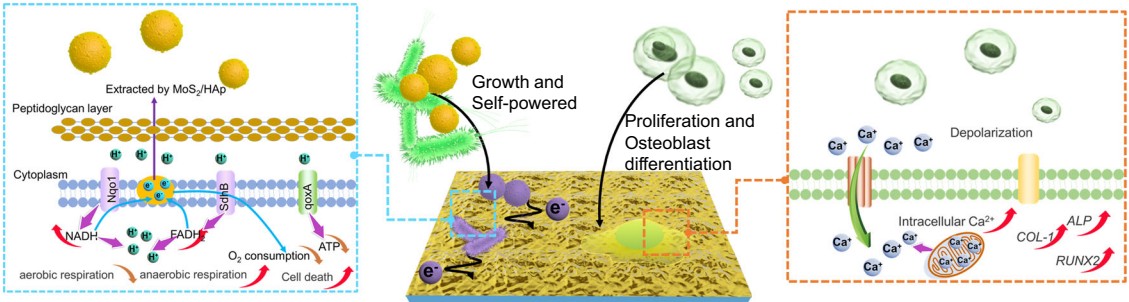

**Fig. 1 Design of the bacteria or MSCs–HA/MoS$_2$–Ti6 biohybrid system to prevent bacterial infection and improve bone-tissue regeneration simultaneously.** Electrons extracting from bacteria to HA/MoS$_2$-Ti6 lead to the metabolism pathway changes (left), and the potential of HA/MoS$_2$-Ti6 leads to the osteogenic differentiation of MSCs due to upregulated Ca$^{2+}$ level (right).

and the 0.60 nm lattice spacing of the neighboring lattice fringes represented the (002) plane of MoS$_2$[27], whereas the 0.34 nm lattice spacing matched the (002) plane of HA[28], manifesting the formation of HA/MoS$_2$ heterojunction with compact interface contact via the process of laser cladding and CVD. Simultaneously, it was obvious that the formed HA/MoS$_2$ heterojunction was in a solid–melt state, which was observed by its high-angle annular dark-field (HAADF) image (Supplementary Fig. 6a). The corresponding elemental mapping images of HA/MoS$_2$ showed a homogeneous distribution of Ca, P, Mo, and S elements in the coating (as shown in Supplementary Fig. 6b–6e). The ultraviolet-visible-near infrared (UV-Vis-NIR) absorption spectra of Ti6, HA–Ti6, MoS$_2$–Ti6, and HA/MoS$_2$–Ti6 were shown in Fig. 2g. Compared with Ti6 and HA–Ti6, both MoS$_2$–Ti6 and HA/MoS$_2$–Ti6 showed a widened visible light absorption spectrum, especially for the latter. According to the Kubelka–Munk function, the bandgap of MoS$_2$–Ti6 and HA/MoS$_2$–Ti6 was 1.73 eV and 1.38 eV, respectively (Fig. 2h). Ultraviolet photoelectron spectroscopy (UPS) spectra of MoS$_2$–Ti6, and HA/MoS$_2$–Ti6 showed that the binding energy of the secondary cutoff edge of MoS$_2$–Ti6 and HA/MoS$_2$–Ti6 was 16.43 eV and 16.50 eV, respectively. The work function of MoS$_2$–Ti6 and HA/MoS$_2$–Ti6 was 4.77 eV and 4.70 eV, respectively (Fig. 2i, j). The conduction band (CB) minimum for MoS$_2$–Ti6 and HA/MoS$_2$–Ti6 was −4.19 eV and −4.20 eV, respectively, and the corresponding valence band (VB) minimum was −5.92 eV and −5.58 eV, respectively. The bacteria owned the biological redox potential (BRP) due to the disulfide bonds on the bacterial membrane, which ranged from −4.12 eV to −4.84 eV[29,30]. Furthermore, the potential of HA/MoS$_2$–Ti6 was lower than the BRP, ensuring the smooth transfer of electrons from membrane proteins to HA/MoS$_2$–Ti6(Fig. 2k)[31,32]. The interaction between HA/MoS$_2$–Ti6 and bacteria was further analyzed by culturing bacteria on different substrates (Fig. 2l and Supplementary Fig. 7). Compared with the number of bacterial colonies in Ti6 group (both S. aureus and E. coli), both HA–Ti6 and MoS$_2$–Ti6 groups had similar bacterial counts. In comparison with Ti6 group, HA–Ti6 group showed about 0.09-log and 0.06-log reduction in bacterial counts against S. aureus and E. coli, respectively. MoS$_2$-Ti6 group displayed 0.16-log and 0.2-log reduction in bacterial counts against S. aureus and E. coli, which was possibly ascribed to the layered structure of MoS$_2$[13,17]. In contrast, there were only a few bacterial colonies in HA/MoS$_2$–Ti group, which showed 1.81-log and 1.88-log reduction in bacterial counts against S. aureus and E. coli, respectively, suggesting its great broad-spectrum antibacterial performance. Besides, the methicillin-resistant *Staphylococcus aureus* (MRSA) treated with HA/MoS$_2$–Ti6 showed 2.36-log reduction in bacterial counts (Supplementary Fig. 8). The result suggested that HA/MoS$_2$–Ti6 group also had great antibacterial ability against MRSA. The antibacterial performances of both Ti6

and HA/MoS$_2$–Ti6 with lower bacterial concentrations were further assessed by spread plate (Supplementary Fig. 9). The bacterial counts in log (CFU mL$^{-1}$) in HA/MoS$_2$–Ti6 group were reduced from 7.32-log to 5.22-log after 6 h culturing with an initial bacterial concentration of 10$^5$ CFU mL$^{-1}$, and from 8.66-log to 6.26-log after 6 h cultivation with an initial concentration of 10$^6$ CFU mL$^{-1}$. But the one in Ti6 group almost did not change. The antibacterial activity of HA/MoS$_2$ was further assessed by minimum inhibitory concentration (MIC) and minimum bactericidal concentration (MBC) (Supplementary Fig. 10 and Supplementary Fig. 11). The values of MIC and MBC of HA/MoS$_2$–Ti6 were about 5 mg mL$^{-1}$ and 10 mg mL$^{-1}$.

**Activation of bacterial death by HA/MoS$_2$–Ti6.** To further evaluate the antibacterial properties of different samples, bacterial live (green)/dead (red) staining and bacterial morphologies examination were also performed. As shown in Fig. 3a, compared to Ti6 group, there were many live bacteria and little dead bacteria in the two groups of HA–Ti6 and MoS$_2$–Ti6. In contrast, HA/MoS$_2$–Ti6 group showed few live bacteria, suggesting its great antibacterial ability. In addition, SEM was used to examine the morphologies and membrane integrity of bacteria on the surface after co-incubation with different samples for 6 h (S. aureus) and 12 h (E. coli), respectively. The morphologies of S. aureus were spherical and intact in both Ti6 and HA–Ti6 groups. A small number of S. aureus were shrunk in MoS$_2$–Ti6 group (indicated by red arrows), but most of bacteria retained the intact shape. For HA/MoS$_2$–Ti6 group, the cell membranes of S. aureus were irregular or completely damaged. Similarly, E. coli in both Ti6 and HA–Ti6 groups showed normal morphologies, and only a small number of bacteria were damaged on MoS$_2$–Ti6 group. However, E. coli on HA/MoS$_2$–Ti6 group became corrugated and distorted or even partly broken, further demonstrating the best antibacterial efficacy of HA/MoS$_2$–Ti6 among these samples. In addition, the antibacterial mechanism of HA/MoS$_2$–Ti6 against S. aureus was further analyzed in detail (Supplementary Fig. 5, Supplementary Fig. 10–12 and Figs. 2i and 3a). First, the values of MIC and MBC of HA/MoS$_2$–Ti6 were about 5 mg mL$^{-1}$ and 10 mg mL$^{-1}$, respectively. It is known that the antibacterial agents are considered bactericidal if the value of MBC is <4 times the value of MIC[13]. The ratio of MBC to MIC was 2, indicating that HA/MoS$_2$–Ti6 had a certain bacteriostatic effect (Supplementary Fig. 10-11). Second, the influence of HA/MoS$_2$-Ti6 on bacterial morphology was further assessed by SEM. The bacterial membranes were irregular or completely broken after culturing on the surface of HA/MoS$_2$-Ti6 for 6 h, suggesting the bactericidal ability of HA/MoS$_2$-Ti6 (Fig. 3a). Next, the influence of antifouling ability on antibacterial activity was evaluated. Generally, good hydrophilicity is beneficial for developing an antifouling surface. In this work, as shown in Supplementary Fig. 5, the HA/MoS$_2$-

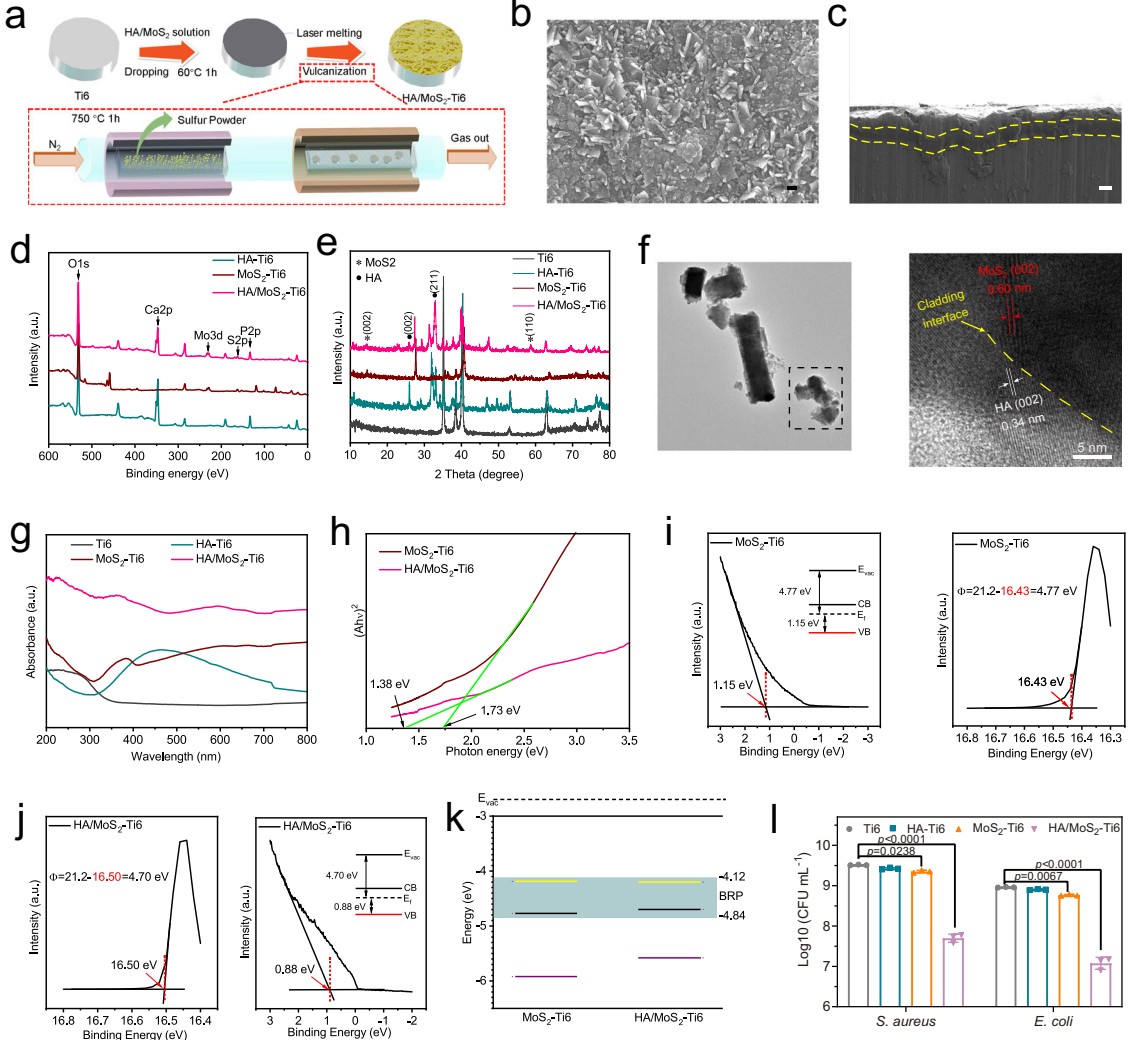

**Fig. 2 Characterization and antibacterial performance of HA/MoS₂-Ti6. a** A schematic illustration of the synthesis process of HA/MoS₂-Ti6. **b** SEM image of HA/MoS₂-Ti6 ($n = 3$ independent samples) (scale bar = 200 nm). **c** Cross-section image of HA/MoS₂-Ti6 ($n = 3$ independent samples) (scale bar = 10 μm). **d** XPS survey scan of HA–Ti6, MoS₂-Ti6, and HA/MoS₂-Ti6. **e** XRD patterns of Ti6, HA–Ti6, MoS₂-Ti6, and HA/MoS₂-Ti6. **f** TEM image of HA/MoS₂ scraped from HA/MoS₂-Ti6 (left) (scale bar = 500 nm) and corresponding HRTEM image (right). ($n = 3$ independent samples). **g** UV-Vis-NIR spectra of Ti6, HA–Ti6, MoS₂-Ti6, and HA/MoS₂-Ti6. **h** The corresponding plot analysis of optical bandgap of pristine MoS₂ and HA/MoS₂. **i** UPS spectra measured by He I (hν = 21.22 eV) spectra. of the MoS₂-Ti6. The valence band of MoS₂-Ti6 (left). The secondary electron cutoff of MoS₂-Ti6 (right). **j** UPS spectra measured by He I (hν = 21.22 eV) spectra. The secondary electron cutoff of HA/MoS₂-Ti6 (left). The valence band of HA/MoS₂-Ti6 (right). **k** Energy level diagram of HA/MoS₂-Ti6 and the representation of electron transfer mechanism to bacterial membrane. Red line, green line, and blue line stands for CB level, VB level, and Fermi level, respectively. And the dark blue area shows the range of BRP (from −4.12 to −4.84 eV). **l** The antibacterial results of Ti6, HA–Ti6, MoS₂-Ti6, and HA/MoS₂-Ti6 against *S. aureus* and *E. coli*. Data represented as mean ± standard deviations from a representative experiment. Error bar represents the standard deviation. $p$-values were generated by two-way analysis of variance (ANOVA) with Dunnett's multiple comparison test. ($n = 3$ independent samples). Source data are provided as a Source Data file.

Ti6 owned a good hydrophilic surface because of many hydroxyl groups on the surface, and similarly, both HA–Ti6 and MoS₂-Ti6 had good hydrophilicity. However, the former exhibited highly effective antibacterial efficacy while the latter two had no antibacterial effect (Fig. 2l). These results suggested that the antibacterial ability of HA/MoS₂-Ti6 was not caused by the antifouling effect. Additionally, the laser cladding induced structure on Ti6 (SLM-Ti6) almost had no antibacterial effect (Supplementary Fig. 12), suggesting the antibacterial ability of HA/MoS₂-Ti6 was not caused by the laser-induced structure. All in all, the antibacterial activity predominantly resulted from the bactericidal ability of HA/MoS₂-Ti6.

RNA sequencing analysis was then used to clarify the differences in gene expression of *S. aureus* cultured on HA/

MoS₂–Ti6 and Ti6. As shown in Supplementary Fig. 13a, there were 54 upregulated and 47 downregulated genes in HA/MoS₂–Ti6 group compared to Ti6 group. The principal component analysis indicated that the points between Ti6 and HA/MoS₂–Ti6 had a long distance, which revealed a great gene expression difference between Ti6 group and HA/MoS₂–Ti6 group (Supplementary Fig. 13b). As shown in Fig. 3b, c, among the gene ontology (GO) enrichment analysis, the downregulated genes were mainly enriched in the ATP metabolic process, cytochrome-*c* oxidase activity, and ATP-binding cassette transporter complex. The upregulated genes were mainly enriched in plasma membrane respiratory chain complex II, anaerobic electron transport chain, and anaerobic respiration. Bacterial metabolism was altered to adapt to the external environment.

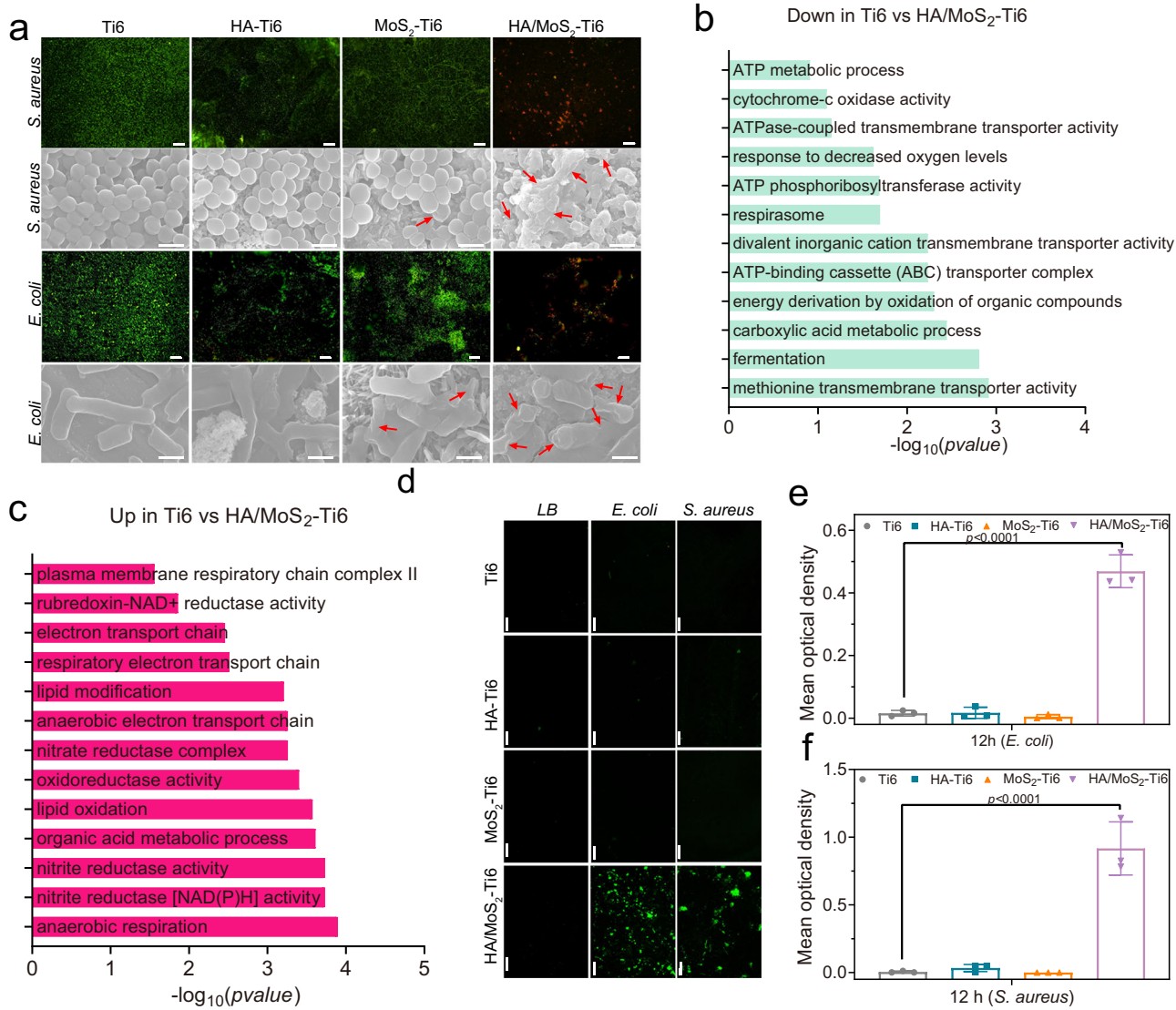

**Fig. 3 Activation of bacterial death by HA/MoS₂-Ti6. a** Fluorescent images of stained bacteria (scale bar = 50 μm) and FE-SEM morphologies (scale bar = 1 μm) after treatment on various surface for *S. aureus* and *E. coli*. **b** Downregulated gene ontology (GO) enrichment analysis in HA/MoS₂-Ti6 compared with Ti6. **c** Upregulated GO enrichment analysis in HA/MoS₂-Ti6 compared with Ti6. **d** Detection of ROS production induced by different specimen of Ti6 HA-Ti6, MoS₂-Ti6, and HA/MoS₂-Ti6 (scale bar = 50 μm). **e**, **f** Quantitative analysis of fluorescent intensity. **e**, **f** Data represented as mean ± standard deviations from a representative experiment. Error bar represents the standard deviation. **b**, **c**, **e**, **f** *p*-values were generated by one-way ANOVA with Dunnett's multiple comparison test. (*n* = 3 independent samples). Source data are provided as a Source Data file.

Since intracellular ROS production is the marker of the cell redox equilibrium[33], in order to further demonstrate the influence of HA/MoS₂–Ti6 on bacterial metabolism, 2,7-dichlorodihydro-fluorescein diacetate (DCFH-DA) was used to detect ROS in *S. aureus* and *E. coli*. As shown in Fig. 3d, after incubation with *S. aureus* and *E. coli* for 6 and 12 h, respectively, obvious green fluorescence was observed in HA/MoS₂-Ti6 group, indicating the production of ROS in both *S. aureus* and *E. coli* after culturing on the surface of HA/MoS₂–Ti6. In contrast, there was no ROS production in the two kinds of bacteria growth on the surface of Ti6, HA–Ti6, and MoS₂–Ti6. Those results showed that bacteria could effectively produce intracellular ROS when they adhered to the surface of HA/MoS₂–Ti6. As illustrated in Fig. 3e, f, the quantitative analysis of fluorescent intensity was further used to analyze the content of ROS in different groups. Compared to the fluorescent intensity of Ti6 group, the one of HA/MoS₂–Ti6 group was significantly stronger, whereas that of HA–Ti6 and MoS₂–Ti6 almost had no difference. Those suggested that the

state of bacterial redox equilibrium was disturbed when bacteria were in contact with HA/MoS₂–Ti6.

**HA/MoS₂-Ti6 extracts electrons from bacteria to alter the metabolism.** To further study the mechanism of electron transfer from *S. aureus* to the surface of HA/MoS₂-Ti6, a heat map was used to analyze the difference in gene expression between the two groups of HA/MoS₂-Ti6 and Ti6 (Fig. 4a). The expression levels of *narG*, *fadA*, and *sdhB* genes were upregulated, which was related to anaerobic respiration. The expression levels of *qoxA*, *sdhA*, and *hisB* genes were downregulated, which was related to aerobic respiration. To further clarify the effect of HA/MoS₂–Ti6 on bacterial electron transfer, the linear sweep voltammetry (LSV) curves of bacteria with different samples were measured[34]. As shown in Supplementary Fig. 14, when there were no bacteria on the sample surface, there was no significant difference in the saturated current of MoS₂–Ti6 and HA/MoS₂–Ti6. However, when MoS₂–Ti6 and HA/MoS₂–Ti6 were in contact with *S.*

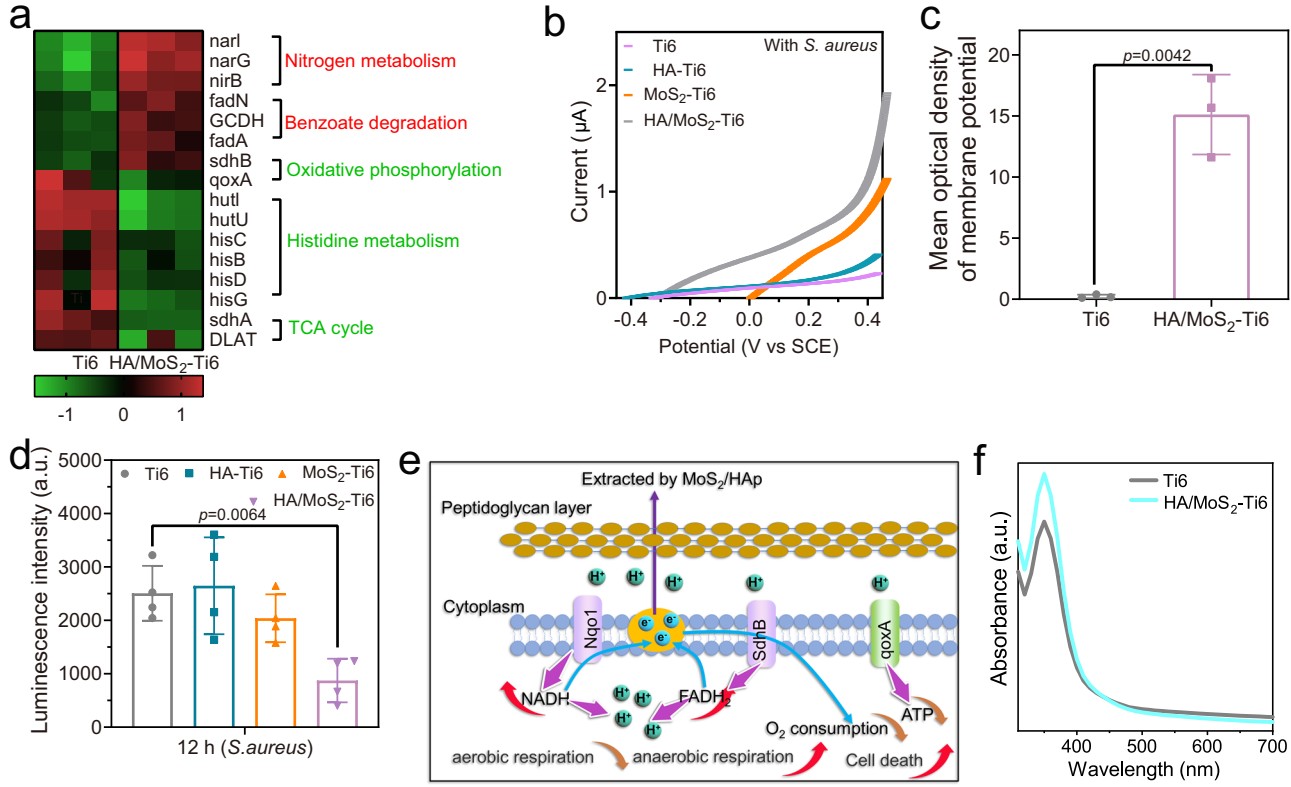

**Fig. 4 HA/MoS₂-Ti6 extracted electrons from bacteria. a** Expression change of the nitrogen metabolism, benzoate degradation, oxidative phosphorylation, histidine metabolism, and TCA cycle between Ti6 and HA/MoS₂-Ti6. **b** LSV curves of different samples without bacteria (*S. aureus*). **c** Mean optical density of bacterial membrane potential of Ti6 and HA/MoS₂-Ti6. ($n = 3$ independent samples). **d** ATP level of *S. aureus* adhered on different samples for 12 h in the dark presenting the bacterial metabolic activity intensity. ($n = 4$ independent samples). **e** Complete process of *S. aureus* activated by HA/MoS₂-Ti6 for further stimulating bacterial death. **f** Oxygen concentration of *S. aureus* with different treatment presenting the bacterial metabolic state. **c, d** Data represented as mean ± standard deviations from a representative experiment. Error bar represents the standard deviation. *p*-values were analyzed by one-way ANOVA with Dunnett's multiple comparison test. Source data are provided as a Source Data file.

*aureus*, HA/MoS₂–Ti6 group showed a larger saturated current (Fig. 4b). It was inferred that when *S. aureus* was in contact with HA/MoS₂–Ti6, the electrons of *S. aureus* moved to the surface of HA/MoS₂–Ti6 through the built-in electric field at the bacteria-semiconductor interface. The DiBAC₄(3) was used to measure cell membrane potential[35]. And the fluorescence intensity of HA/MoS₂–Ti6 group was higher than that of Ti6 group, indicating that the electrons from the cell membrane were transferred to the surface of HA/MoS₂–Ti6, which altered the membrane potential of *S. aureus* (Fig. 4c). Cell membrane potential depolarization was caused by the interaction between HA/MoS₂ coating and bacteria. The zeta potential was used to measure the bacterial membrane potential, and the bacterial membrane potential could reflect the inherent metabolic state of the bacteria[36]. In addition, the bacterial surface potential was determined by the charges of molecular on bacterial membrane, which was in correlation with the metabolic process that happened in cells aerobically and anaerobically cultured[37]. As shown in Supplementary Fig. 15, when bacteria were cultured on the surface of HA/MoS₂-Ti6, the zeta potential turned more negative compared with bacteria cultured on the surface of Ti6. It was ascribed to the change of metabolism from aerobic respiratory to anaerobic respiratory. Furthermore, the ATP activity of *S. aureus* after different treatments was further analyzed (Fig. 4d). ATP is an energy molecule in bacteria that participates in some bacterial physiological activities, such as respiration[38]. When cells become necrotic or apoptotic, the ATP levels in bacteria will be declined. In this work, after bacteria adhered to the surface of HA/MoS₂–Ti6, the bacteria in this

group exhibited the lowest ATP level among all groups, which further indicated that HA/MoS₂-Ti6 effectively destroyed the normal respiration of bacteria. The complete process of electron transfer from *S. aureus* to HA/MoS₂–Ti6 was schematically illustrated in Fig. 4e. First, when bacteria adhered to the coating, the typical proteins in bacterial membrane, including nicotinamide adenine dinucleotide (NAD) + hydrogen (H) (NADH) quinone oxidoreductase 1 (Nqo1), succinate dehydrogenase (SdhB), and quinol oxidase subunit qoxA, which participated in the transfer of electrons during bacterial respiration, contacted the material. Next, Nqo1 and SdhB drove the formation of NADH and flavin adenine dinucleotide (FAD) + 2hydrogen (H) (FADH2). Meanwhile, the downregulated quoxA inhibited O₂ consumption and ATP formation. The remaining electrons were transferred to HA/MoS₂–Ti6 through membrane cytochromes or electrically conductive pili because of the potential difference between bacteria and HA/MoS₂–Ti6[39,40]. The above results indicated that when *S. aureus* adhered to HA/MoS₂–Ti6, its metabolism pathway was altered from aerobic to anaerobic respiration. These changes disrupted the bacterial redox equilibrium, leading to cell death. Figure 4f showed that Ti6 group consumed more oxygen than HA/MoS₂-Ti6 group. The result demonstrated that bacteria were induced into anaerobic respiration when bacteria adhered to the surface of HA/MoS₂-Ti6. Furthermore, the antibacterial mechanism of HA/MoS₂-Ti6 was further verified by the antibacterial activity of a kind of strictly aerobic bacterium. Pseudomonas aeruginosa (P. aeruginosa) was a kind of strictly aerobic bacterium, which could not adapt to anaerobic environments. As

shown in Supplementary Fig. 16, HA/MoS$_2$-Ti6 group almost had no antibacterial ability against *P. aeruginosa*, suggesting that the antibacterial mechanism of HA/MoS$_2$-Ti6 was related to the change of bacterial metabolism pathway.

**Osteogenic differentiation of MSCs.** Another part of the study was analysis of osteogenic ability of HA/MoS$_2$–Ti6. Therefore, the influence of the potential of HA/MoS$_2$–Ti6 on stem cells was further analyzed. The influence of the potential of HA/MoS$_2$-Ti6 group on cell membrane potential was first examined. As shown in Fig. 5a, b, the DiBAC$_4$(3) fluorescence intensity of HA/MoS$_2$–Ti6 group was higher than that of Ti6 group, indicating the transfer of electrons from the cell membrane to HA/MoS$_2$–Ti6 group, which altered the membrane potential of MSCs. The cellular redox potential was regulated via redox couples during the cellular metabolic process, and its cellular redox potential was from $-4.12$ eV to $-4.84$ eV. HA/MoS$_2$–Ti6 had oxidizing substances, which could disturb redox equilibrium and increase the content of intracellular ROS. The immediate response of Nrf2-mediated antioxidant defense was triggered due to the redox disequilibrium[41]. Mitochondria play a vital role in the generation of metabolic energy, and there is electrons transfer during ATP formation[42,43]. In this work, mitochondrial membrane potential was used to judge whether the defense was successful. As shown in Fig. 5c, the ratio of the fluorescence intensities of J-monomers and J-aggregates was decreased, indicating that the mitochondrial membrane potential was increased, which induced the disturbance of the mitochondrial function and the enhancement of the respiration[44]. Furthermore, the fluorescence imaging results revealed that the intracellular Ca$^{2+}$ level in HA/MoS$_2$–Ti6 group was much higher than that in Ti6 group (Supplementary Fig. 17). The high level Ca$^{2+}$ was transported into mitochondria, which regulated the mitochondrial metabolism and caused transient depolarization of mitochondrial membrane. This fluctuation of Ca$^{2+}$ from the endoplasmic reticulum played an important physiological role for cell proliferation[45,46]. Furthermore, a high concentration of intracellular Ca$^{2+}$ suggested a great ability for osteogenic differentation[47]. DAPI (4′,6-diamidino-2-phenylindole) and tetramethylrhodamine (TRITC)-conjugated phalloidin (actin) were used to characterize the morphologies of MSCs in the presence of different samples (Fig. 5d). It is well known that MSCs with spread shape and larger mean radius ratio suggest good biocompatibility and high potential for osteogenic differentiation of materials[48]. As shown in Fig. 5e, f, compared to Ti6 group, the cell viability in HA/MoS$_2$–Ti6 group was greatly increased after culturing for 1 day, 3 days, and 7 days, respectively. The 3-(4,5-dimethylthiazol-2-yl)-2,5-diphenyltetrazolium bromide (MTT) assay after culturing on the surface of Ti6, SLM-Ti6, and HA/MoS$_2$-Ti6 was further performed (Supplementary Fig. 18). The cell viability in SLM-Ti6 group was almost the same as that in Ti6 group. In contrast, the cell viability in HA/MoS$_2$-Ti6 group was obviously enhanced. These results suggested that the cell proliferation was mainly due to the enhanced mitochondrial metabolism rather than three-dimensional structure caused by laser cladding. Additionally, the three kinds of samples including HA–Ti6, MoS$_2$–Ti6, and HA/MoS$_2$–Ti6 had great biocompatibility. Alkaline phosphatase (ALP) activity was used to measure the ability of osteogenic differentiation in the presence of different samples. As depicted in Fig. 5g, the ALP activity of both HA–Ti6 and HA/MoS$_2$-Ti6 groups was higher than that of Ti6 and MoS$_2$–Ti6 groups. This was related to the osteoconductive HA and the potential of HA/MoS$_2$-Ti6, which activated the Wnt/β-catenin and Wnt/Ca$^{2+}$ pathways. The osteogenic genes were further analyzed via real-time quantitative reverse transcription PCR (RT-qPCR). Compared with Ti6 group, HA/MoS$_2$–Ti6

group exhibited higher expression level of *ALP*, runt-related transcription factor 2 (*RUNX2*), and type I collagen (*COL-I*) (Fig. 5h). Extracellular mineralization could be defined as the calcium deposition of samples, which were stained by alizarin red. As shown in Fig. 5i, j, compared to Ti6 group, both HA–Ti6 and HA/MoS$_2$–Ti6 groups presented highly enhanced matrix mineralization.

**Bone integration of HA/MoS$_2$-Ti6 in vivo.** According to the above results, the antibacterial and osteogenic properties of HA/MoS$_2$–Ti6 were outstanding in vitro, so it was chosen for in vivo research. The implants were placed into the tibia of male rats (Supplementary Fig. 19a and Supplementary Fig. 19b, red arrow). The implant position was shown by an X-ray image (Supplementary Fig. 19c, red arrow). The inflammatory response and remaining bacteria in the bone tissue around the implant were evaluated via hematoxylin and eosin (H&E) and Giemsa staining after 14 days of implantation (Fig. 6a, b)[49]. H&E staining was used to observe inflammatory cells, for example, neutrophils. And there were many neutrophils around the implant of Ti6 with *S. aureus* group (Fig. 6a, blue arrow). On the contrary, the number of neutrophils in HA/MoS$_2$-Ti6 with *S. aureus* group was less than that in Ti6 with *S. aureus* group because of the effective antibacterial efficacy of the composite coating. Without adding *S. aureus* initially, both Ti6 and HA/MoS$_2$-Ti6 groups had fewer neutrophils. Giemsa staining (Fig. 6b, black arrow) was used to observe the number of bacteria around the tissue of the implant. Without adding *S. aureus* initially, both Ti6 and HA/MoS$_2$-Ti6 groups had no bacteria. In contrast, once adding *S. aureus* initially, few bacteria in HA/MoS$_2$-Ti6 group were observed after 14 days while there were many bacteria in Ti6 group, further proving the highly effective antibacterial efficacy of HA/MoS$_2$–Ti6 group in vivo. As a result, the initially added *S. aureus* caused a much weaker inflammation in the implant of HA/MoS$_2$-Ti6 but a strong inflammation reaction in Ti6 group. The Giemsa staining further indicated that HA/MoS$_2$-Ti6 killed the most bacteria with few bacteria remaining. Consequently, it was hard to form the biofilm on the surface of HA/MoS$_2$-Ti6. The bacterial spread plate and colonies were further used to quantify the antibacterial ability of different samples in vivo (Fig. 6c, d). Compared to the group of Ti6 + *S. aureus*, the bacteria in HA/MoS$_2$-Ti6 + *S. aureus* group were 2.65-log reduction in bacterial counts. Furthermore, the bone regeneration ability of different samples was assessed via microcomputed tomography (micro-CT) after 4 weeks of implantation[50]. Three cylindrical regions around the implant with a diameter of 2.51 mm and a thickness of 0.40 mm were selected, and three- and two-dimensional images were reconstructed with a special software program (Fig. 6e). The newly formed bone tissues (Obj. V [object volume]/TV [tissue volume]) were used to analyze the bone mass. HA/MoS$_2$-Ti6 + *S. aureus* and HA/MoS$_2$–Ti6 without *S. aureus* groups showed newly formed bone tissues of 44.62% and 51.67%, respectively, much more than the corresponding one of Ti6 + *S. aureus* group (25.40%) and Ti6 without *S. aureus* group (35.13%) (Fig. 6f). Both methylene blue-acid magenta staining and Safranin-O/Fast Green staining were used to assess the histopathological conditions around the implant after 4 weeks implantation (Fig. 6g, h). As for methylene blue-acid magenta staining, the red color represented the mineralized bone tissue in methylene blue-acid fuchsin staining. Compared to Ti6 + *S. aureus* group, HA/MoS$_2$–Ti6 + *S. aureus* group, HA/MoS$_2$–Ti6 without *S. aureus* group, and Ti6 without *S. aureus* group showed better bone-implant contact. The quantitative results showed that HA/MoS$_2$–Ti6 + *S. aureus* group and HA/MoS$_2$–Ti6 without *S. aureus* group had excellent bone contact of 80.15% and 89.58%, respectively, which was much

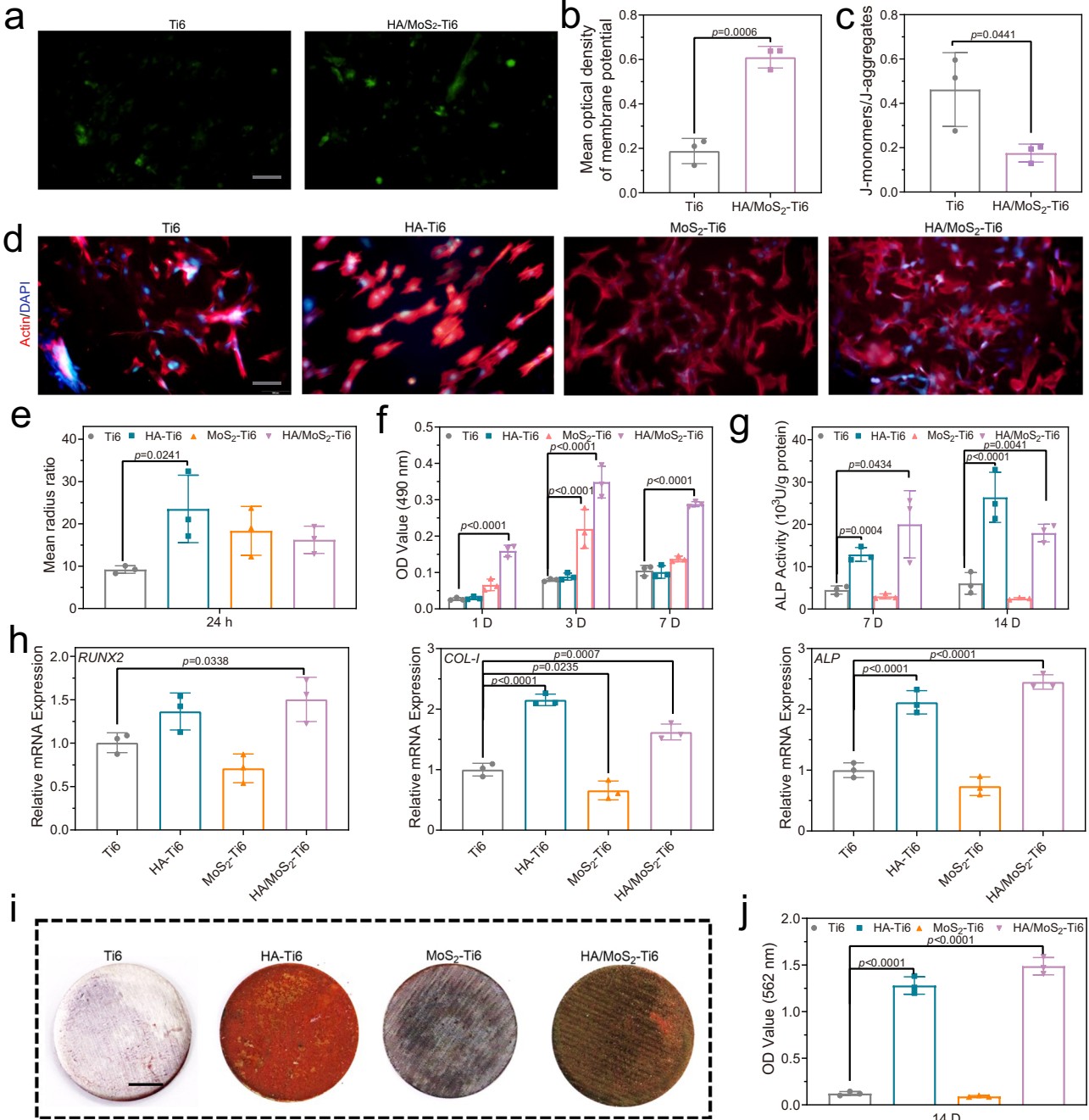

**Fig. 5 Osteoblastic differentiation of MSCs by osteoconductive HA and the innate potential of HA/MoS₂ heterojunction. a** Fluorescence imaging of cell membrane potential, scale bar = 100 μm. **b** Quantitative analysis of fluorescent intensity of the cell membrane potential. **c** Quantitative analysis of fluorescent intensity of the mitochondria membrane potential. **d** Fluorescence pictures of Ti6, HA–Ti6, MoS₂-Ti6, and HA/MoS₂-Ti6 after culturing for 24 h. The red color represented cell actin, and the blue color indicated the cell nucleus. Scale bar = 100 μm. **e** Mean radius ratio of cells in the fluorescence pictures. **f** The cell viability after culturing for 1 day, 3 days, and 7 days. **g** The ALP activity after culturing for 3 days, 7 days, and 14 days. **h** Osteogenic-related gene expression of runt-related transcription factor 2 (*RUNX2*), *ALP*, and type I collagen (*COL-I*) after 14 days. **i** Alizarin red staining of different samples after 14 days, scale bar = 0.5 cm. **j** Quantification of the Alizarin red staining intensity. **b**, **c**, **e**, **h**, **j**, **f**, **g** Data represented as mean ± standard deviations from a representative experiment. Error bar represents the standard deviation. (*n* = 3 independent samples). **b**, **c**, **e**, **h**, **j** *p*-values were generated by one-way ANOVA with Dunnett's multiple comparison test. **f**, **g** *p* values were generated by two-way ANOVA with Dunnett's multiple comparison test. Source data are provided as a Source Data file.

higher than the corresponding value of the Ti6 without *S. aureus* group (44.18%) and Ti6 + *S. aureus* group (12.12%). As regards Safranin-O/Fast Green staining, the osteogenesis was represented by green color while cartilage was marked by red or orange color.

As illustrated in Fig. 6i, j, HA/MoS₂–Ti6 without *S. aureus* group and HA/MoS₂-Ti6 + *S. aureus* group contained many osteoblasts and no chondrocytes in the bone tissues. In contrast, Ti6 + *S. aureus* group contained many chondrocytes while Ti6 without *S.

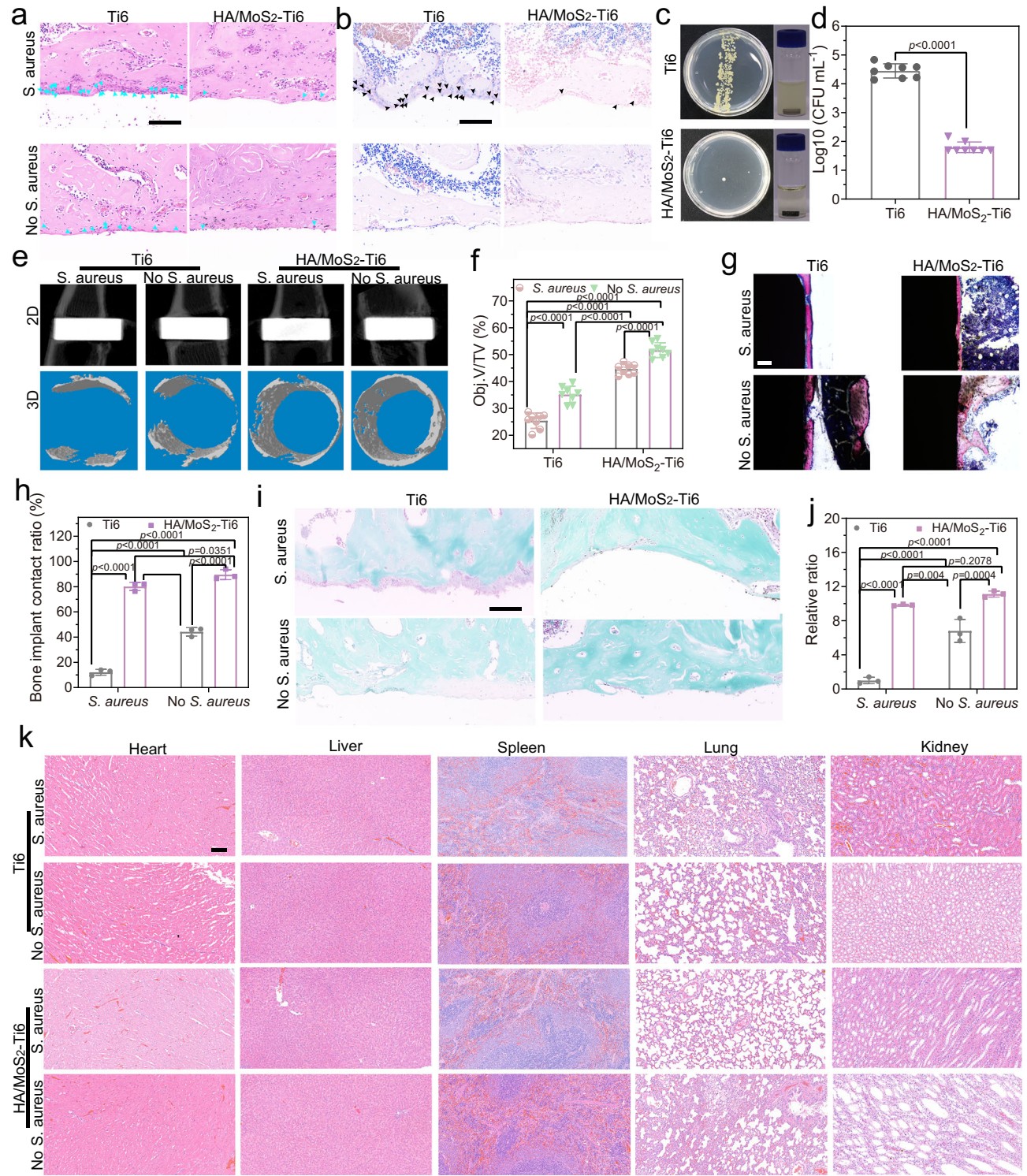

aureus group contained some chondrocytes. Compared with Ti6 + S. aureus group, the quantitative results of osteogenesis indicated that the relative ratio of HA/MoS$_2$–Ti6 without S. aureus group, HA/MoS$_2$–Ti6 + S. aureus group, and Ti6 without S. aureus group was 11.13, 9.84, and 6.8, respectively, compared with Ti6 with S. aureus group. The in vivo toxicology of major organs including the heart, liver, spleen, lung, and kidney was detected by H&E staining after 4 weeks. As shown in Fig. 6k, no signs of organ damage were found, indicating that there was no obvious histological toxicology of prepared materials. These results revealed that the synthesized HA/MoS$_2$–Ti6 not only had

the self-antibacterial performance but also possessed good osteoconductive, which made the coating of HA/MoS$_2$ prepared by laser cladding be an effective strategy to solve the two major concerns of bacterial infection and osseointegration on artificial implants simultaneously.

In summary, the synthesized HA/MoS$_2$ coating by laser cladding took advantage of the difference of energy metabolism between MSCs and bacteria to kill pathogenic bacteria and promote the bone regeneration on Ti implants simultaneously. On the one hand, HA/MoS$_2$ coating induced bacteria death via the transfer of electrons from the adhered bacteria to the surface of HA/MoS$_2$ semiconductor.

**Fig. 6 Bone integration of HA/MoS$_2$-Ti6 in vivo. a** H&E staining of bone tissues around the implants under different conditions ($n = 8$ independent samples), scale bar = 100 μm. **b** Giemsa staining of bone tissues around the implants ($n = 8$ independent samples), scale bar = 100 μm. **c** In vivo antibacterial evaluation, (Left) the remaining bacteria colonies on Ti6 and HA/MoS$_2$-Ti6 rods pulled out from the tissues after implantation for 14 days, (Right) the optic pictures of bacteria extracted from the implants of Ti6 and HA/MoS$_2$-Ti6 after culturing for 24 h. **d** Antibacterial ability of HA/MoS$_2$-Ti6 compared with Ti6 in vivo. ($n = 8$ independent samples). **e** The 2D and 3D pictures of HA/MoS$_2$-Ti6 and Ti6 groups measured by Micro-CT. **f** Bone volume (BV)/tissue volume (TV) values of HA/MoS$_2$-Ti6 and Ti6 groups. ($n = 8$ independent samples). **g** Methylene blue-acid magenta staining of the newly formed bone tissues on the bone-implant interface, scale bar = 100 μm. **h** Bone area ratios of samples calculated from the methylene blue-acid magenta staining. ($n = 3$ independent samples). **i** Safranin-O/Fast Green staining. The green color is osteogenesis, and the red or orange color is cartilage. The scale bar is 100 μm. **j** Histomorphometric measurements of osteogenesis. The area was taken from 20 μm around the implant. The control was Ti6 + *S. aureus* group. ($n = 3$ independent samples). **k** Biological assessment of heart, liver, spleen, lung, and kidney ($n = 8$). The scale bar is 100 μm. **d**, **f**, **h**, **j** Data represented as mean ± standard deviations from a representative experiment. Error bar represents the standard deviation. *p*-values were generated by two-way ANOVA with Dunnett's multiple comparison test. Source data are provided as a Source Data file.

---

Differential gene expression analysis revealed that after bacteria adhered to HA/MoS$_2$-Ti6, the coating of HA/MoS$_2$ not only altered the gene expression of bacteria in response to the changes of electron receptors but also modulated their metabolism pathway from aerobic to anaerobic respiration. On the other hand, HA/MoS$_2$ coating also enhanced the osteogenic proliferation of MSCs by altering the potentials of cell membrane and mitochondrial membrane. Hence, this work provides a promising strategy to design multi-functional bone implants with highly effective antibacterial activity and osteogenic ability simultaneously based on the potential difference between specific coating materials and microbes as well as MSCs.

## Methods

**Chemicals.** Sodium chloride (NaCl), potassium chloride (KCl), dimethyl sulfoxide (DMSO), ethylene glycol((CH$_2$OH)$_2$), and sodium citrate were from Sinopharm Chemical Reagent Co., Ltd. L-ascorbic acid, MTT were purchased from Sigma Chemical Co. (St. Louis, MO, USA). Trypsin-EDTA, penicillin and streptomycin, TRITC-conjugated phalloidin (actin), and 4′, 6-diamidino-2-phenylindole (DAPI) were from yeasen. Dulbecco's modified eagle medium (DMEM) and fetal bovine serum (FBS) were from Gibco. The alkaline phosphatase assay kit (AKP Microplate test kit) was purchased from Nanjing JianCheng Bioengineering Institute. BCA Protein Assay kit goat serum was purchased from Solarbio (Beijing, China). 2′,7′-dichlorofluorescein diacetate (DCFH-DA) was purchased from the Beyotime Institute of Biotechnology. (Shanghai, China). The total cell RNA kit, total bacteria RNA kit, and total DNA kit were from Omega. PrimeScript RT Master Mix and 2×SYBR Premix Ex Taq II were from TaKaRa.

**Synthesis of different samples.** First, the Ti6Al4V plates (Ti6) were polished *via* different grits (240, 400, and 800) of silicon carbide (SiC) sandpapers. Next, the polished plates were washed with absolute ethanol and deionized water for 15 min, respectively. HA (10 mg mL$^{-1}$), MoS$_2$ (10 mg mL$^{-1}$), and HA/MoS$_2$ solutions were dropped on the Ti6 surface before vigorous ultrasonication for half an hour. The ratio (w/w) of HA and MoS$_2$ in the HA/MoS$_2$ solution was 50:50, which were called the precoated samples. Following this, the different precoated samples were treated by laser cladding before being dried in an oven at 60 °C for 1 h.

Regarding the laser cladding process, the precoated samples were placed on the holder of the instrument (JHM-1GY-300B; Lumonics). With a wavelength of 1.06 μm, laser cladding was carried out by CW 2 kW Nd: YAG laser, with optimal parameters of laser current = 90 A, pulse width = 2 ms, frequency = 20 Hz, spot diameter = 0.6 mm, and scanning speed = 5 mm s$^{-1}$. This was the process of preparation of HA–Ti6. After laser cladding, both MoS$_2$–Ti6 and HA/MoS$_2$–Ti6 samples were sulfurated using the CVD method. Specifically, 0.3 g of sulfur powder and the samples were put in a quartz tube filled with nitrogen at 1 atm pressure. Then, the tube furnace was heated to 750 °C at a heating rate of 10 °C min$^{-1}$ and maintained for 1 h, and then naturally cooled down to room temperature. This was the process of preparation of MoS$_2$–Ti6 and HA/MoS$_2$–Ti6.

**Characterization of different samples.** SEM images were obtained using a JSM-6510LV and a JEM-2100F microscope (JEOL, Tokyo, Japan). TEM images were recorded by a Talos F200x electron microscope (FEI Co., USA). XRD (D8A25; Bruker, Germany) was used to determine the crystal structure of the samples. XPS (ESCALAB 250Xi; Thermo Scientific, USA) was employed to disclose the surface elemental compositions of the samples. UPS was measured using an ESCALAB 250Xi instrument with a monochromatic He I light source (21.2 eV). An InVia reflex system (Renishaw) operating at 532 nm was employed to obtain Raman spectra. The JC2000D Contact Angle System (POWEREACH, China) was used to measure the water contact angles of the samples at room temperature. A UV-Vis-

NIR spectrometer (UV-3600; Shimadzu, Tokyo, Japan) was used to obtain the spectra of samples.

**Calculation of CB and VB of MoS$_2$-Ti6 and HA/MoS$_2$-Ti6.** The values of CB minimums and VB minimums were obtained through combination of UV-Vis-NIR spectra and UPS spectra. UV-Vis-NIR spectroscopy was used to get the bandgap of material. UPS was used to determine the highest occupied molecular orbital (HOMO) from the Fermi level. Then, we could the predict lowest unoccupied molecular orbital (LUMO) from the observed bandgap. A complete expression of the calculation process of CB and VB was descripted as following:

The work function (φ) could be calculated using Eq. (1): $\varphi = h\nu - E_{SEO}$. Here, $h\nu = 21.20$ eV, which represented the energy of the monochromatic ionizing light, while $E_{SEO}$ was the secondary electron onset, which was obtained from the linear extrapolation of the UPS spectrum.

The Fermi level ($E_F$) was obtained from the work function using Eq. (2): $E_F = -\varphi$.

The position of the VB maximum ($E_{VB}$) was obtained from Eq. (3): $E_{VB} = E_F - X$, in which X was obtained from the extrapolation of the onsets in the UPS spectrum.

The CB minimum potential ($E_{CB}$) was obtained from Eq. (4): $E_{CB} = E_{VB} + E_{BG} = E_F - X + E_{BG}$. Here, the bandgap energy $E_{BG}$ was obtained by Tauc plots.

The CB position of MoS$_2$–Ti6 and HA/MoS$_2$–Ti6 was determined by the UPS spectra. The work function of MoS$_2$–Ti6 and HA/MoS$_2$–Ti6 was calculated to be 4.77 and 4.70 eV, respectively, by using the method of a linear approximation to the UPS spectra. The Fermi level of MoS$_2$–Ti6 and HA/MoS$_2$–Ti6 was calculated to be −4.77 and −4.70 eV, respectively. Next, the $E_{VB}$ level of MoS$_2$–Ti6 and HA/MoS$_2$–Ti6 was −5.92 and −5.58 eV separately. The average bandgap energy value was 1.73 eV for MoS$_2$–Ti6 and 1.38 eV for HA/MoS$_2$–Ti6, which was obtained from the Tauc plots. According to Eq. (4), the calculated $E_{CB}$ level of MoS$_2$–Ti6 and HA/MoS$_2$–Ti6 was −4.19 and −4.20 eV, respectively.

**Culturing of bacteria.** Gram-positive methicillin-susceptible *S. aureus* (ATCC 25923), Gram-positive methicillin-resistant *S. aureus* (MRSA) (43300), Gram-negative *E. coli* (ATCC 8099), and Gram-negative *P. aeruginosa* (ATCC 15692) were cultured in a sterile Luria–Bertani (LB) medium (10 g L$^{-1}$ of back to-tryptone, 10 g L$^{-1}$ of NaCl, and 5 g L$^{-1}$ of bacto-yeast extract). The bacterial counts were obtained from the spread plate of different samples. *P. aeruginosa* was a strictly aerobic Gram-negative bacterium[51].

**In vitro antibacterial experiment.** The antimicrobial efficiency of Ti6, HA–Ti6, MoS$_2$–Ti6, and HA/MoS$_2$–Ti6 against *E. coli* methicillin-susceptible *S. aureus* ATCC 25923, MRSA, and *P. aeruginosa* was evaluated by the spread plate method. The samples were sterilized before experiments. The 96-well plates were used to place the specimens separately, and then 200 μL of bacterial suspension (both *E. coli*, *S. aureus*, MRSA, and *P. aeruginosa*) with $2.5 \times 10^7$ CFU mL$^{-1}$ was added into each well, followed by incubation at 37 °C for 6 h (*S. aureus*), 6 h (MRSA), 6 h or 12 h (*P. aeruginosa*) or 12 h (*E. coli*) in an orbital shaker at 200 r.p.m. Ti6 group was used as the control group, while HA–Ti6, MoS$_2$–Ti6, and HA/MoS$_2$–Ti6 groups were separately used as the experimental groups. After incubation, bacterial suspension was diluted 40,000 times for *S. aureus* and 10,000 times for *E. coli* with phosphate-buffered saline (PBS). Following this, 20 μL of bacterial suspension was coated on standard LB agar plates and cultivated in a furnace at 37 °C for 24 h to count the number of colonies on the plate. In addition, the LB medium was centrifuged at 12,000 × g for 20 min for further texting oxygen content.

To further illustrate the antibacterial activity, SEM was used to qualitatively examine the bacteria. The samples were fixed with 2.5% glutaraldehyde for 2 h after incubation, and then washed three times with PBS, subsequently dehydrated in turn with different concentrations of ethanol (10, 30, 50, 70, 90, and 100 v/v%) for

15 min each, and then freeze-dried. The bacterial morphologies of different groups were observed by SEM.

For live/dead staining, the samples with *S. aureus* and *E. coli* were incubated at 37 °C for 12 h. Then, they were soaked in blended dyes (live/dead baclight bacterial viability kit) in the dark for 15 min, followed by rinsing with PBS. Finally, pictures were taken with a fluorescent microscope (IX73; Olympus, Tokyo, Japan).

For the measurement of MIC, the MIC values of these nanomaterials were obtained using a series of diluted samples. A dilution series of HA/MoS$_2$ (0 mg mL$^{-1}$, 0.3125 mg mL$^{-1}$, 0.625 mg mL$^{-1}$, 1.25 mg mL$^{-1}$, 2.5 mg mL$^{-1}$, 5 mg mL$^{-1}$, 10 mg mL$^{-1}$, 20 mg mL$^{-1}$, and 40 mg mL$^{-1}$) were prepared, in which the steps were identical to the process of synthesis of HA/MoS$_2$-Ti6. *S. aureus* was diluted to $1.7 \times 10^7$ cells mL$^{-1}$ with LB broth, and 200 μL of the diluted solution was added into the surface of different samples in the 96-well plate, followed by incubation at 37 °C for 6 h. Optical density (OD) was obtained with a microplate reader.

For the measurement of MBC, the MBC values of these nanomaterials were obtained using a series of diluted samples. After the same antibacterial process as MIC, the bacterial suspension was diluted 60,000 times, and 20 μL bacterial suspension was coated on LB agar plates and cultured at 37 °C for 24 h. Finally, MBC was obtained from the number of colonies on the plate.

**RNA sequence for *S. aureus***. *S. aureus* were cultured with Ti6 and HA/MoS$_2$–Ti6 for 6 h and then *S. aureus* were collected to extract the total RNA using TRIzol reagent (Invitrogen, CA, USA). RNA sequencing was performed via the HiSeq 4000 SBS Kit (300 cycles; Illumina, CA, USA). Data analysis was performed by FastqStat.jar (v0.11.4) and RSeQC (v2.6.4). Gene Ontology (http://www.geneontology.org) and Kyoto Encyclopedia of Genes and Genome (http://www.genome.jp/kegg/) were used to analyze the gene functions. Differential gene expression analysis was performed using the R package edgeR (v3.24), and those genes conformed to |log2FC| > 1 (*p*-value < 0.05) were considered to be differentially expressed genes.

**Detection of ROS**. Fluorescence imaging was performed to investigate the intracellular ROS level. In brief, each sample (including Ti6, HA–Ti6, MoS$_2$–Ti6, and HA/MoS$_2$–Ti6) was placed in 96-well plates and filled with 200 μL of bacterial suspension ($5 \times 10^7 \sim 10^8$ CFU mL$^{-1}$ in LB medium or pure LB medium), followed by incubation at 37 °C for 12 h for *E. coli* and 6 h for *S. aureus* in an orbital shaker at 200 r.p.m. Next, the medium was removed and the samples were washed twice with PBS. Next, the cells on the sample surface were stained for 30 min in the dark by 200 μL DCFH-DA (10 μM), and then the samples were washed twice with PBS to remove the excess dye. In the following step, an inverted fluorescent microscope (IFM; IX73) was used to observe the samples as mentioned above.

**Cell potential of MSCs and *S. aureus***. DiBAC4(3) was used to assess the cell membrane potential of *S. aureus* and MSCs. As for MSCs, 10$^4$ cells per well were seeded on Ti6 and HA/MoS$_2$–Ti6 surfaces. After culturing for 24 h, the cell growth medium was discarded and the new growth medium with DiBAC4(3) (2 μM) was added and cultured for 30 min at 37 °C. Following this, the cell potential was assessed via fluorescence images that were obtained via an IFM (570 Olympus, IX73). Concerning *S. aureus*, $5 \times 10^7$ CFU mL$^{-1}$ *S. aureus* were added to the Ti6 and HA/MoS$_2$–Ti6 surfaces. After culturing for 6 h, the LB medium was discarded and transferred into a new LB medium with DiBAC4(3) (5 μM) for 30 min at 37 °C. Subsequently, the bacterial potential was assessed via fluorescence images that were obtained through 570 Olympus (IX73).

**Cell culture**. MSCs were obtained from Tongji Hospital (Wuhan, China). The cells were cultured in a growth medium (minimum essential medium eagle alpha modification: fetal bovine serum: antibiotics penicillin/streptomycin [100 U/mL] = 89:10:1 [v/v]) at 37 °C in a 5% CO$_2$ environment.

**Mitochondrial membrane potential of MSCs**. A mitochondrial membrane potential assay kit (JC-1; SBJbio life sciences, Nanjing, China) was used to obtain the mitochondrial membrane potential of MSCs. First, 10$^4$ cells per well were seeded on Ti6 and HA/MoS$_2$–Ti6 surfaces. After this, the mitochondrial membrane potential of MSCs was evaluated via the instruction manual of JC-1.

**Ca$^{2+}$ fluorescence imaging of MSCs**. 10$^4$ cells per well were seeded on Ti6 and HA/MoS$_2$–Ti6 surfaces and incubated for 24 h. Next, Ca$^{2+}$ indicator (Flu-3 AM, 5 μM) was added into the fresh medium. And then, the cells were washed with Hanks' Balanced Salt Solution (HBSS) three times. Finally, the pictures were obtained with IFM (Olympus, IX73) at 488 nm.

**Cell proliferation assays**. Cell proliferation was evaluated by 3-(4,5-dimethylthiazol-2-yl)-2,5-diphenyltetrazolium bromide (MTT). Briefly, MSCs (10$^5$ cells mL$^{-1}$) were seeded on the surface of different samples and cultured for 1, 3, and 7 days. Following this, the MTT solution (0.5 mg mL$^{-1}$) was added to each well and incubated for 4 h to form purple precipitates. Finally, the optical density (OD) of the liquid was tested at 490 nm with a microplate reader after dissolving it with dimethyl sulfoxide.

**Osteoblastic differentiation assays**. The cell osteoblastic differentiation assays were assessed via ALP, RT-qPCR, and Alizarin Red S (ARS) staining assays. The cells were cultured in an osteogenesis-inducing medium (growth medium with 10 mM β-glycerol phosphate, $10 \times 10^{-9}$ M dexamethasone, and 50 μg mL$^{-1}$ L-ascorbic acid). As for the ALP assay, the cells were lysed with 1% Triton X-100 at 37 °C for 2 h after culturing for 7 and 14 days. Finally, ALP activity was evaluated via an ALP assay kit. In addition, a BCA protein assay kit (Solarbio, China) was used to detect the protein content to obtain the ALP activity per unit protein. Regarding RT-qPCR, the cells were collected to extract the total RNA using a total RNA kit after culturing for 14 days. Following this, the total RNA was converted into cDNA through the PrimeScript RT Master Mix. Finally, 2×SYBR Premix Ex Taq II was used to perform RT-qPCR with cDNA using the CFX Connect real-time system. As for ARS staining, the cells were fixed with 4% formaldehyde for 20 min after culturing for 14 days. Next, the cells were stained with 2% Alizarin Red (pH 4.2) for 10 min. Finally, cetylpyridinium chloride in 10% w/v 10 mM sodium phosphate (pH 7.0) was used for further quantitative analysis at 562 nm with a microplate reader.

**In vivo biological evaluation**. Sprague Dawley rats (300–350 g) were bought from the Beijing Huafukang Bioscience Cojnc. The study was performed following the Guide for the Care and Use of Laboratory Animals of the National Institutes of Health. The ethical aspects of the animal experiments were approved by the Animal Ethical and Welfare Committee (AEWC) of the Institute of Radiation Medicine, Chinese Academy of Medical Sciences (Approval No. YSY-DWLL-2021016). The 64 male rats were separated into eight sections (*n* = 8 per section), including (1) Ti6 with *S. aureus* after culturing 2 weeks, (2) Ti6 with no *S. aureus* after culturing 2 weeks, (3) HA/MoS$_2$–Ti6 with *S. aureus* after culturing 2 weeks, (4) HA/MoS$_2$–Ti6 with no *S. aureus* after culturing 2 weeks, (5) Ti6 with *S. aureus* after culturing 4 weeks, (6) Ti6 with no *S. aureus* after culturing 4 weeks, (7) HA/MoS$_2$–Ti6 with *S. aureus* after culturing 4 weeks, and (8) HA/MoS$_2$–Ti6 with no *S. aureus* after culturing 4 weeks. As for *S. aureus* infection group, bacterial suspension (20 μL) with a density of 10$^7$ CFU mL$^{-1}$ *S. aureus* was uniformly coated on the surface of Ti6 and HA/MoS$_2$–Ti6 rods before use. For surgery, pentobarbital sodium salt solution (30 mg kg$^{-1}$, 1% w/w) was used to anesthetize the rats by injection. Then, the different samples were implanted into the tibia near the knee joint. The rats were fed in the same way, and after 2–4 weeks, they were euthanized via an overdose of chloral hydrate.

**Spread plate analysis and histological analysis**. To determine the antibacterial efficiency of Ti6 and HA/MoS$_2$–Ti6 rods, the rods (Ti with *S. aureus* and HA/MoS$_2$–Ti6 with *S. aureus*) were removed after 2 weeks, rolled on a standard agar plate for four rounds, and then cultured for 24 h at 37 °C. The rods after rolling on a standard plate were plated in small glass vials and then cultured for 24 h at 37 °C. The bacterial colonies and glass vials were photographed on a rolling trace using a digital camera. Meanwhile, H&E and Giemsa staining were used to determine the bacterial contamination of bone-tissue and bone implants after 2 weeks. The H&E and Giemsa staining of samples without bacteria were used as the control group, which was used to assess the influence of bacteria on bone tissue. A fluorescence microscope was used to analyze the histopathological microtomography.

**Bone micro-CT and histopathological evaluation**. A micro-CT system (USDA Grand Forks Human Nutrition Research Center, Grand Forks, ND, USA) was used to analyze the quantification of gross bone morphology and microarchitecture. To determine the newly formed bone around the implants, the bone volume for each total sample volume (BV/TV) was calculated.

Simultaneously, Safranin-O/Fast Green staining was used to process the samples, which was used to evaluate the osteogenic or chondrogenic differentiation on the surface of implants. The osteogenesis was represented by green while the cartilage was marked by red or orange. The osteogenesis ratio defined the percentage of osteogenesis with a region extending 20 μm from the implant surface. Methylene blue-acid fuchsin staining was used to analyze the mineralized bone tissue (red) around the implant/bone interface. The images were obtained using a fluorescence microscope.

**Statistical analysis**. All the quantitative data were analyzed by the *t*-test, one-way ANOVA with Dunnett's multiple comparison test, or two-way ANOVA with Dunnett's multiple comparison test. And the data were presented as mean with s.d. Values of *$p < 0.05$, **$p < 0.01$, ***$p < 0.001$ and ****$p < 0.0001$ were considered statistically significant.

**Reporting summary**. Further information on research design is available in the Nature Research Reporting Summary linked to this article.

## Data availability

The RNA-seq data generated in this study are available on the National Center for Biotechnology Information (NCBI) database under the BioProject PRJNA777398. Source data are provided with this paper.

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

## Acknowledgements

This work is jointly supported by the China National Funds for Distinguished Young Scientists (no. 51925104) and the National Natural Science Foundation of China (nos. 51871162, 52173251 and 51771069).

## Author contributions

These authors contributed equally: J.F. and W.Z. These authors conceived the concept: J.F., W.Z., X.L. and S.W. These authors jointly supervised this work: J.F., W.Z., X.L., S.W. and C.L. These authors synthesized the samples and conducted the materials characterizations: J.F., W.Z., X.L. and C.L. These authors provided important experimental insights: Y.Z., Z.C., Y.L., S.Z., Z.L. and D.Z.

## Competing interests

The authors declare no competing interests.
