## [Peer Review File · Nature Communications]

Self-activating anti-infection implantREVIEWER COMMENTS

Reviewer #1 (Remarks to the Author):

The manuscript demonstrates a MoS₂ and HA modified implant designed to prevent *S. aureus* and *E. coli* infection and accelerate bone integration simultaneously. The results are carefully analyzed and convincing. However, a few points need to be addressed by the authors before acceptance:

1. The meaning of "Self-powered" was still not clear after reading whole manuscript. More explanation about "Self-powered" should be added.
2. High temperature (~750°C) are used to form HA/MoS₂-Ti6. Many researches have reported that vacancies were formed in MoS₂ by heating. Did the structure of MoS₂ change in this experiment? More characterization and explanation are necessary.
3. How do authors get values of CB minimums and VB minimums? More explanation should be added in the manuscript.
4. Except for DiBAC4(3) fluorescence intensity, specific values of cell membrane potential should be measured via cell membrane potential assay kit or other methods.
5. More appropriate references should be cited in the process of electron transfer.
6. "The Nrf2-mediated antioxidant defense was triggered to restore the redox disequilibrium." It is not clear how Nrf2-mediated antioxidant defense was triggered.
7. "the mitochondria function was disturbed, which indicates enhanced respiration, increased cell proliferation," Not sure if the authors can conclude that the disturbed mitochondria function increased cell proliferation. The authors should explain better before jumping into such conclusion.
8. "The enhanced cell viability in the presence of HA/MoS₂-Ti6 group might be due to the enhanced mitochondrial metabolism and three-dimensional structure caused by laser cladding." What is the relation between cell viability and 3D structure? More data and explanation should be added.

Reviewer #2 (Remarks to the Author):

Fu et al. reported the preparation of MoS₂ and hydroxyapatite modified Ti implant and the application of the same for antibacterial activity and accelerate bone integration. Even though, the study is in detail and interesting but limited towards the novelty of the work, both in respect to material and application (cited below). Moreover, this report is lacking proper explanation/citation of structure and activity of layered MoS₂ in antimicrobial activity which is already well established (Nat. Nanotechnol. 2016, 11 (12), 1098-1104. and J. Am. Chem. Soc. 2018, 140, 12634.).

Considering lacking novelty in this work, I would not recommend this manuscript for publication in nature communications and doubt this work meet the high requirement of this journal.

Detailed comments were included below:

1. There is nothing called 'Transition metal haloalkane'. It should be Transition metal dichalcogenide.
2. "Also, nano-MoS₂ facilitates the separation of electron-hole pairs, reducing the distance between electrons and hole diffusion to the material surface." This is not correct statement. Authors need to look at this and must include the proper description of direct bandgap of nanoMoS₂. Also the reference 17, related to this statement is inappropriate.
3. The ref. 12 related to 'MoS₂ to be widely used in biomaterials' is too less information to describe the importance of MoS₂ in biological application. There are few recent good reviews are there which need to be cited here. For an example, Small, 2019,15, 1803706. etc.
4. Preparation of MoS₂/HA composite was reported earlier as well as applied in few biological applications such as cell adhesion, cell proliferation, , implantation, osteogenesis etc. (ACS

Biomater. Sci. Eng. 2019, 5, 4511–4521). So the claim about 'However, little is known about the biological role of MoS₂/HA in bacteria, especially in terms of extracellular electron transfer to MoS₂/HA coating' is not correct.

5. How the UV-Vis-NIR samples was prepared? Based on the description, they are not dispersible. Moreover, the MoS₂ or composite of MoS₂ generally exhibit characteristic peaks at ~675 and ~610 nm, 458 and 405 nm related to direct excitonic transition at the K point, which is not observed in the reported spectra.

6. In antibacterial activity assay, what quantity of material used, was not mentioned. In other word, what is the MIC/MBC for these nanomaterials? How effective these nanomaterials in compare to other reported systems in compare to both antibacterial activity and osteogenesis?

7. Why the HA-MoS₂ composite is more effective that HA or MoS₂ alone?

Reviewer #3 (Remarks to the Author):

In this manuscript, the authors introduce a composite material of hydroxyapatite (HA) and molybdenum sulfide (MoS₂). They hypothesize that this material may be both osteoinductive (promoting differentiation of mesenchymal stem cells into osteoblasts) and antibacterial (through interrupting extracellular aerobic respiration of bacterial pathogens). They set forth to demonstrate this with both in vitro studies (including *Staphylococcus aureus* and *E coli* as representative gram positive and gram negative pathogens) as well as a rat model of a contaminated tibial defect (*Staphylococcus aureus*).

The authors show in vitro evidence that supports their hypothesis in general (antimicrobial effect through interruption of metabolic pathways, genetic markers associated with osteoblastic differentiation). They show some in vivo evidence of reduced bacterial burden and additional bone growth in an infected defect model although a control arm (surgical defect without infection using the same biomaterials) was not used.

The manuscript has significant grammatical errors.

Overall, this is an exciting new direction for the field of device infection. Using metabolism-based therapies rather than traditional antibiotics should circumvent traditional mechanisms of antimicrobial resistance. The in vitro studies performed are thorough; however, I am concerned that the authors have overstated some of their results (for example, expressing reduction of bacteria as a percentage rather than absolute log). In addition, I am concerned that proper controls were not used in the animal model and it is not clear to me if n=6 is properly powered. Overall though, I appreciate the authors' attention to detail and their choice of assays, including genetic analysis.

Specific comments below are as follows:

1. Page 2, line 28. What does self-powered mean? Please clarify.
2. Page 3, line 70. Is molybdenum sulfide truly widely used in biomaterials? I would change this language as this is likely untrue.
3. Page 4, line 81. This sentence appears incomplete.
4. Page 4, line 83. This is likely not the first time a biomaterial has been proposed that is both osteoconductive and antibacterial, although it is still unclear what "self-powered" implies.
5. Page 6, line 142. "Coculturing" typically implies culture of two different species or lines; I would suggest that this is culture on a substrate.
6. The phrase "antibacterial efficiency" is used often when describing this material. The mechanism is not clear to me whether the change in bacterial colonies is due to an anti-adhesive effect versus a bacteriostatic or bacteriocidal effect.
7. In Figure 1L, it is shown that while statistically significant in effect and a two-log difference in

concentration, there still remained between 7-8 log Staph aureus and E coli on the most promising material surface (HA/MoS2-Ti6). This is far from sterilizing and may clinically still lead to infection. It is misleading to express this difference as a percentage change, I recommend representing it as a log change given bacterial growth kinetics. Please comment.

8. The results on page 7 and 8 would be even more exciting if the experiment was replicated with Staph aureus mutants/knock outs that lack molecular mechanisms to adapt to anaerobic activity to corroborate the sequencing data.

9. Page 12, line 291. It is microcomputed tomography, not microtomography.

10. Page 12, line 305. Chondrocytes can be precursors to bony development (endochondral ossification) so would be careful about implying that appearance of chondrocytes is not consistent with osteogenesis.

11. Page 19, line 478. Please specify if this isolate of Staphylococcus aureus is methicillin-susceptible or -resistant.

12. Similar to my above comment on in vitro results, while there was ~two fold log reduction in presence of Staphylococcus aureus on the in vitro work, there still remains $\sim 10^4$ bacteria (see Fig 5c). Expressing this reduction as a percentage is misleading given bacterial growth kinetics. While it is reassuring that there was greater bone growth in the treatment arm, it is difficult to know if the remaining 10^4 bacteria in the long term would compromise the surgical site. In addition, proper matching controls were not performed- there should be two additional groups (Ti6 with no bacterial inoculation and HA/MoS2-Ti6 with no bacterial inoculation). It is unclear if $n=6$ per group was correctly powered.

Response to Reviewer 1#

Original Comment: The manuscript demonstrates a MoS₂ and HA modified implant designed to prevent *S. aureus* and *E. coli* infection and accelerate bone integration simultaneously. The results are carefully analyzed and convincing. However, a few points need to be addressed by the authors before acceptance:

Reply: Thank you so much for positive comment about “The results are carefully analyzed and convincing”.

Comment 1: The meaning of “Self-powered” was still not clear after reading whole manuscript. More explanation about “Self-powered” should be added.

Reply: Thank you so much for your professional suggestions. We were so sorry for our unclear expression of self-powered. Self-powered systems were often used to describe systems which did not need an outside energy supply to work (Nature Nanotechnology 2010, 5(5), 366–373; Advanced Functional Materials 2008, 18(22), 3553-3567; Nano Today 2010, 5(6), 512-514). In this work, MoS₂ and HA modified implant took advantage of the energy from metabolic process of *S. aureus* and *E. coli* to active antibacterial process, which did not need extra energy. We thought that the process belonged to self-powered process. We added the explanation about “self-powered” at **page 4 line 21**: “When bacteria were contacted with HA/MoS₂ coating on Ti6, the potential difference between HA/MoS₂-Ti6 and *S. aureus* made the electrons from microbial membranes transfer to HA/MoS₂ coating surface. The process took advantage of the energy from metabolic process of *S. aureus* and *E. coli* without extra energy to trigger antibacterial activity of HA/MoS₂-Ti6. It was a self-powered process of anti-infection.”

Comment 2: High temperature (~750°C) are used to form HA/MoS₂-Ti6. Many researches have reported that vacancies were formed in MoS₂ by heating. Did the structure of MoS₂ change in this experiment? More characterization and explanation are necessary.

Reply: Thank you so much for your suggestions. The structure of MoS₂ does not change in the experiment. We have performed electron paramagnetic resonance (EPR) experiment to measure sulfur vacancies in HA/MoS₂-Ti6, and there were no signal of vacancies (Supplementary Fig. 4). In addition, we added the related content at **page 6 line 4**: “The electron paramagnetic resonance (EPR) experiment was further performed to characterize the vacancies in HA/MoS₂-Ti6 (**Supplementary Fig. 4**). HA/MoS₂-Ti6 had no signal of vacancies compared with Ti6, suggesting that the structure of MoS₂ did not change in this experiment. Sulfur powder would be sublimed under high temperature, which the sulfur powder microenvironment limited the formation of vacancies.”

Supplementary Fig. 4

Comment 3: How do authors get values of CB minimums and VB minimums? More explanation should be added in the manuscript.

Reply: Thank you so much for your suggestions. We added detail steps for getting values of CB minimums and VB minimums at **page 23 line 16**: “The values of CB minimums and VB minimums were obtained through combination of ultraviolet-visible (UV-Vis) spectroscopy and ultraviolet photoemission spectra (UPS). UV-Vis spectroscopy was used to get band gap of material. UPS was used to determine the highest occupied molecular orbital (HOMO) from the fermi level. Then, we could predict lowest unoccupied molecular orbital (LUMO) from the observed bandgap. A complete expression of the calculation process of CB and VB was described as following:

The work function (ϕ) can be calculated using Eq. (1): $\phi = h\nu - E_{\text{SE0}}$. Here, $h\nu = 21.20$ eV, represents the energy of the monochromatic ionizing light, while E_{SE0} is the secondary electron onset, obtained from the linear extrapolation of the UPS spectrum.

The Fermi level (E_{F}) is obtained from the work function using Eq. (2): $E_{\text{F}} = -\phi$.

The position of the valence band maximum (E_{VB}) is obtained from Eq. (3): $E_{\text{VB}} = E_{\text{F}} - X$, in which X is obtained from the extrapolation of the onsets in the UPS spectrum.

The conduction band minimum potential (E_{CB}) is obtained from Eq. (4): $E_{\text{CB}} = E_{\text{VB}} + E_{\text{BG}} = E_{\text{F}} - X + E_{\text{BG}}$. Here, the bandgap energy E_{BG} is obtained by Tauc plots.

The CB position of $\text{MoS}_2\text{-Ti6}$ and $\text{HA/MoS}_2\text{-Ti6}$ are determined by the UPS spectra. The work function of $\text{MoS}_2\text{-Ti6}$ and $\text{HA/MoS}_2\text{-Ti6}$ was estimated to be 4.77 eV and 4.70 eV, applying the method of a linear approximation to the UPS spectra. The Fermi level of $\text{MoS}_2\text{-Ti6}$ and $\text{HA/MoS}_2\text{-Ti6}$ was estimated to be -4.77 eV and -4.70 eV. Next, the E_{VB} level of $\text{MoS}_2\text{-Ti6}$ and $\text{HA/MoS}_2\text{-Ti6}$ was estimated to be -5.92 eV and -5.58 eV. The average band gap energy value (1.73 eV for $\text{MoS}_2\text{-Ti6}$, 1.38 eV for $\text{HA/MoS}_2\text{-Ti6}$) obtained from the Tauc plots. The E_{CB} level of $\text{MoS}_2\text{-Ti6}$ and $\text{HA/MoS}_2\text{-Ti6}$ was -4.19 and -4.20 eV, respectively, through Eq. (4).”

Comment 4: Except for DiBAC4(3) fluorescence intensity, specific values of cell membrane potential should be measured via cell membrane potential assay kit or other methods.

Reply: Thank you so much for your suggestions. We performed zeta potential to measure the specific values of cell membrane potential. And we added at **page 12 line 23**: “The zeta potential measurement was used to measure the bacterial membrane potential, which reflected the inherent metabolic state of the bacteria.³² In addition, the bacterial surface potential was altered by changing the molecular nature of surface potential, which was in correlation with the metabolic process that happened in cells aerobically and anaerobically cultured.³³ As shown in **Supplementary Fig. 15**, when bacteria were cultured on the surface of $\text{HA/MoS}_2\text{-Ti6}$, the zeta potential would turn more negative compared with bacteria cultured on the surface of Ti6. It was due to the metabolism transfer from aerobic respiratory to anaerobic respiratory.”

- 32 Klodzinska, E. *et al.* Effect of zeta potential value on bacterial behavior during electrophoretic separation. *Electrophoresis* **31**, 1590-1596 (2010).
- 33 Lavaisse, L. M., Hollmann, A., Nazareno, M. A., Disalvo, E. A. Zeta potential changes of *saccharomyces cerevisiae* during fermentative and respiratory cycles. *Colloids Surf., B* **174**, 63-69 (2019).

Supplementary Fig. 15

Comment 5: More appropriate references should be cited in the process of electron transfer.

Reply: Thank you so much for your comment. During the section about process of electron transfer, we have explained that “electron transfer (ET) was necessary process during the cellular metabolism, in which electrons are donated by nicotinamide adenine dinucleotide (reduced form of NADH). Then, the electrons are transferred to a terminal acceptor through a range of redox cofactors located in the bacterial membrane. Furthermore, the potential of HA/MoS₂-Ti6 were lower than those of the redox c-type cytochromes on the bacterial membrane, ensuring that the electrons transferred from membrane proteins to HA/MoS₂-Ti6 through membrane cytochromes or electrically conductive pili.”

Based on your request, have replaced partial references for more plausible explanation about the process of electron transfer. In addition, we also added related explanation in our manuscript at **page 3 line 11**: “Electron transfer is crucial for the energy metabolism of living cells, which fuels most cellular processes. Disturbing the process can stimulate the generation of intracellular reactive oxygen species (ROS) and consequently hinder

their proliferation.⁶ In bacterial respiration, bacteria transfer endogenous electrons to extracellular electron acceptors *via* outer membrane *c*-type cytochromes, conductive bacterial nanowires, and/or self-secreted flavins.^{7,8} and at **page 7 line 9**: “The CB minimums for MoS₂-Ti6 and HA/MoS₂-Ti6 were -4.19 and -4.20 eV, respectively, and their VB minimums were -5.92 and -5.58 eV, respectively. The bacteria had a biological redox potential (BRP) range of -4.12 to -4.84 eV because of the disulfide bonds on the bacterial membrane.^{24,25} Furthermore, the potential of HA/MoS₂-Ti6 were lower than those of the redox *c*-type cytochromes on the bacterial membrane (**Fig. 1k**), ensuring that the electrons transferred from membrane proteins to HA/MoS₂-Ti6.^{26,27}” and at **page 13 line 16**: “The remaining electrons were extracted *via* HA/MoS₂-Ti6 group because of potential differences through membrane cytochromes or electrically conductive pili.^{35,36}”

- 6 Wang, G. *et al.* An antibacterial platform based on capacitive carbon-doped TiO₂ nanotubes after direct or alternating current charging. *Nat. Commun.* **9**, 2055 (2018).
- 7 Shi, L. *et al.* Extracellular electron transfer mechanisms between microorganisms and minerals. *Nat. Rev. Microbiol.* **14**, 651-662 (2016).
- 8 Reguera, G. *et al.* Extracellular electron transfer via microbial nanowires. *Nature* **435**, 1098-1101 (2005).
- 24 Nel, A. E. *et al.* Understanding biophysicochemical interactions at the nano-bio interface. *Nat. Mater.* **8**, 543-557 (2009).
- 25 Baldus, I. B., Gräter, F. Mechanical force can fine-tune redox potentials of disulfide bonds. *Biophys J.* **102**, 622-629 (2012).
- 26 Burello, E. W., A. P. QSAR modeling of nanomaterials. *Wiley Interdiscip. Rev.: Nanomed. Nanobiotechnol.* **3**, 298-306 (2011).
- 27 Burello, E., Worth, A. P. A theoretical framework for predicting the oxidative stress potential of oxide nanoparticles. *Nanotoxicology* **5**, 228-235 (2011).
- 35 Wu, X. *et al.* A role for microbial palladium nanoparticles in extracellular electron transfer. *Angew. Chem. Int. Ed.* **50**, 427-430 (2011).
- 36 Li, J. *et al.* Temperature-responsive tungsten doped vanadium dioxide thin film starves bacteria to death. *Mater. Today* **22**, 35-49 (2019).

Fig. 1k

Comment 6: “The Nrf2-mediated antioxidant defense was triggered to restore the redox disequilibrium.” It is not clear how Nrf2-mediated antioxidant defense was triggered.

Reply: Thank you so much for your comment. In fact, the cellular redox potential was regulated *via* redox couples during cellular process, and its cellular redox potential was at -4.12 to -4.84 eV. The HA/MoS₂-Ti6 had oxidizing substances, which could disturb redox equilibrium. And the content of intracellular ROS was increased. The immediate response of Nrf2-mediated antioxidant defense was triggered to redox disequilibrium. And we added expression at **page 15 line 3**: “The cellular redox potential was regulated *via* redox couples during the cellular process, and its cellular redox potential was at -4.12 to -4.84 eV. The HA/MoS₂-Ti6 had oxidizing substances, which could disturb redox equilibrium. And the content of intracellular ROS was increased. The immediate response of Nrf2-mediated antioxidant defense was triggered to redox disequilibrium.”³⁷

37 Nel, A. *et al.* Nanomaterial toxicity testing in the 21st century: use of a predictive toxicological approach and high-throughput screening. *Acc. Chem. Res.* **46**, 607-621 (2013).

Comment 7: “the mitochondria function was disturbed, which indicates enhanced respiration, increased cell proliferation,” Not sure if the authors can conclude that the disturbed mitochondria function increased cell proliferation. The authors should explain better before jumping into such conclusion.

Reply: Thank you so much for your comment. Based on your suggestion, we added corresponding explanation before jumping into the conclusion. We added the

explanation at **page 15 line 8**: “Mitochondria are the structure of cell metabolism, in which electrons transfer during ATP formation.^{38,39} Mitochondrial membrane potential was used to judge whether the defense was successful. As shown in **Fig. 4c**, the ratio of the fluorescence intensities of J-monomers and J-aggregates was decreased, indicating that the mitochondrial membrane potential was increased and the mitochondria function was disturbed, which indicated enhanced respiration.⁴⁰ Further checking the intracellular Ca^{2+} level using fluorescence (**Supplementary Fig. 17**) revealed that HA/MoS₂-Ti6 group had a higher level than Ti6 group. The Ca^{2+} transported into mitochondria and regulated its metabolism and caused transient depolarization of mitochondrial membrane. This fluctuation of Ca^{2+} from the endoplasmic reticulum played an important physiological role for cell proliferation.^{41,42}”

- 38 Vakifahmetoglu-Norberg, H. *et al.* The role of mitochondria in metabolism and cell death. *Biochem. Biophys. Res. Commun.* **482**, 426-431 (2017).
- 39 Saotome, M. *et al.* Mitochondrial membrane potential modulates regulation of mitochondrial Ca^{2+} in rat ventricular myocytes. *Am. J. Physiol. Heart Circ. Physiol.* **288**, 1820-1828 (2005).
- 40 Inmaculada Martínez-Reyes, Lauren P. D. *et al.* TCA cycle and mitochondrial membrane potential are necessary for diverse biological functions. *Mol. Cell* **61**, 199-209 (2016).
- 41 Upadhyay, R. K. Mitochondrial Ca^{2+} levels lower down rate of metabolic diseases and cardiomyopathies. *J. Stem Cell Res. Ther.* **4**, 82-87 (2018).
- 42 Pinto, M. C. X. *et al.* Calcium signaling and cell proliferation. *Cell. Signalling* **27**, 2139-2149 (2015).

Fig. 4c

Supplementary Fig. 17

Comment 8: “The enhanced cell viability in the presence of HA/MoS₂-Ti6 group might be due to the enhanced mitochondrial metabolism and three-dimensional structure caused by laser cladding.” What is the relation between cell viability and 3D structure? More data and explanation should be added.

Reply: Thank you so much for your suggestions. In our manuscript, the 3D structure caused by laser cladding was not beneficial for cell viability. Based on your request, we performed the MTT assay of different samples (Ti6, SLM-Ti6 and HA/MoS₂-Ti6) for assessing the relation between the cell viability and 3D structure by laser cladding or enhanced mitochondrial metabolism. SLM-Ti6 represented Ti6 that was only treated by selective laser melting, which stood for the three-dimensional structure caused by laser cladding. The cell viability in HA/MoS₂-Ti6 group was great higher than SLM-Ti6 group and Ti6 group, and the cell viability in Ti6 group was similar with SLM-Ti6 group. The results suggested that the enhanced cell viability in the presence of HA/MoS₂-Ti6 group might be due to the enhanced mitochondrial metabolism. We added the explanation at **page 15 line 24**: “The MTT assay after culturing on the Ti6, SLM-Ti6, and HA/MoS₂-Ti6 was further performed at **Supplementary Fig. 18**, and SLM-Ti6 represented Ti6 only treated by selective laser melting. The cell viability in the presence of SLM-Ti6 was almost same as cell viability in presence of Ti6. But the cell viability in the presence of HA/MoS₂-Ti6 was obviously enhanced, suggesting that the cell proliferation was mainly due to the enhanced mitochondrial metabolism rather than three-dimensional structure caused by laser cladding.”

Supplementary Fig. 18

Response to Reviewer 2#

Original Comment: Fu et al. reported the preparation of MoS₂ and hydroxyapatite modified Ti implant and the application of the same for antibacterial activity and accelerate bone integration. Even though, the study is in detail and interesting but limited towards the novelty of the work, both in respect to material and application (cited below).

Reply: Thank you so much for your saying that “the study is in detail and interesting”.

In our manuscript, we proposed novel antibacterial mechanism based on material-bio interface interaction. The electrons of HA/MoS₂ were transferred from the bacteria membrane to HA/MoS₂ after contacting with bacteria due to lower redox potential. Herein, HA/MoS₂ was acting as an electron acceptor. The valence band (VB) maximum and conduction band (CB) minimum for HA/MoS₂ are -4.20 eV and -4.84 eV, respectively. And The bacteria had a biological redox potential (BRP) range of -4.12 to -4.84 eV. As a result, the biohybrid is formed between HA/MoS₂ and bacteria. Furthermore, HA/MoS₂ triggers anaerobic respiration of *S. aureus* and *E. coli* by extracellular electron transfer. RNA sequencing analysis reveals that *S. aureus*'s transcriptional regulations express less aerobic respiration related genes and more anaerobic respiration related genes when interfaced with HA/MoS₂ coating. Therefore, HA/MoS₂ group consumes less oxygen compared with control group. The MoS₂/HA coating presents high-efficiency sterilization rate towards *S. aureus* (98.47%) and *E.*

coli (98.67%) due to induce bacteria into anaerobic respiration.

Moreover, this report is lacking proper explanation/citation of structure and activity of layered MoS₂ in antimicrobial activity which is already well established (Nat. Nanotechnol. 2016, 11 (12), 1098-1104. and J. Am. Chem. Soc. 2018, 140, 12634.).

Reply: Thank you so much for your suggestions. Based on your request, we have added proper explanation/citation of structure and activity of layered MoS₂ in antimicrobial activity. Compared with the previous literatures, MoS₂ were used for photocatalytic antibacterial including the references the reviewer stated (Nat. Nanotechnol. 2016, 11 (12), 1098-1104. and J. Am. Chem. Soc. 2018, 140, 12634.). Because MoS₂ belongs a semiconductor material, it has well visible light or near-infrared photocatalytic response. Therefore, it is often used as a photocatalytic antibacterial material. However, in our manuscript, we used the properties of molybdenum sulfide and hydroxyapatite to construct heterojunctions, and use the material-biological interface to achieve self-powered antibacterial materials. As for bacteria, HA/MoS₂ coating induced bacteria death via bacteria–semiconductor electron transfer. In addition, RNA sequencing revealed that the transcriptional regulations of *S. aureus* express less aerobic respiration–related genes and more anaerobic respiration–related genes when interfaced with HA/MoS₂ coating. The HA/MoS₂ coating presents a high-efficiency sterilization effect toward *S. aureus* and *E. coli* because of bacterial respiration–powered metabolic pathway changes. The coating can enhance the proliferation of mesenchymal stem cells and induce their osteoblastic differentiation by osteoconductive HA and the innate potential of HA/MoS₂ heterojunction.

Corresponding content in our manuscript was shown at **page 7 line 20**: “Whereas MoS₂-Ti6 group displayed 0.16-log and 0.2-log reduction against *S. aureus* and *E. coli* due to its layered structure.^{12,28}”

12 Liu, C. *et al.* Rapid water disinfection using vertically aligned MoS₂ nanofilms and visible light. *Nat. Nanotechnol.* **11**, 1098-1104 (2016).

- 28 Karunakaran, S. *et al.* Simultaneous exfoliation and functionalization of 2H-MoS₂ by thiolated surfactants: applications in enhanced antibacterial activity. *J. Am. Chem. Soc.* **140**, 12634-12644 (2018).

Considering lacking novelty in this work, I would not recommend this manuscript for publication in nature communications and doubt this work meet the high requirement of this journal.

Detailed comments were included below:

Comment 1: There is nothing called ‘Transition metal haloalkane’. It should be Transition metal dichalcogenide.

Reply: Thank you so much for your suggestions. We had revised ‘Transition metal haloalkane’ in our manuscript at **page 3 line 25** into ‘Transition metal dichalcogenide’.

Comment 2: “Also, nano-MoS₂ facilitates the separation of electron–hole pairs, reducing the distance between electrons and hole diffusion to the material surface.” This is not correct statement. Authors need to look at this and must include the proper description of direct bandgap of nanoMoS₂. Also the reference 17, related to this statement is inappropriate.

Reply: Thank you so much for your suggestions. We had revised the statement and reference 17.

We had revised at **page 4 line 1**: “2D MoS₂ nanostructures had a tunable electronic energy state.¹⁴ Also, nanostructured MoS₂ would benefit the separation of electron-hole pairs by decreasing the distances for electrons and holes to diffuse to the surface of the materials and also increasing the reaction sites.¹²”

- 12 Liu, C. *et al.* Rapid water disinfection using vertically aligned MoS₂ nanofilms and visible light. *Nat. Nanotechnol.* **11**, 1098-1104 (2016).

- 14 Lin, Y. C. *et al.* Atomic mechanism of the semiconducting-to-metallic phase transition in single-layered MoS₂. *Nat. Nanotechnol.* **9**, 391-396 (2014).

Comment 3: The ref. 12 related to ‘MoS₂ to be widely used in biomaterials’ is too less information to describe the importance of MoS₂ in biological application. There are few recent good reviews are there which need to be cited here. For an example, Small, 2019,15, 1803706. Etc.

Reply: Thank you so much for your suggestions. We had added literatures in **page 4 line 1**: “And those make it applied in biomaterials.¹³”

13 Yadav, V. *et al.* 2D MoS₂ -based nanomaterials for therapeutic, bioimaging, and biosensing applications. *Small* **15**, e1803706 (2019).

Comment 4: Preparation of MoS₂/HA composite was reported earlier as well as applied in few biological applications such as cell adhesion, cell proliferation, implantation, osteogenesis etc. (ACS Biomater. Sci. Eng. 2019, 5, 4511–4521). So the claim about ‘However, little is known about the biological role of MoS₂/HA in bacteria, especially in terms of extracellular electron transfer to MoS₂/HA coating’ is not correct.

Reply: Thank you so much for your suggestions. We revised it at **page 4 line 7**: “However, little is known about the biological role of HA/MoS₂ in terms of the process of bacterial extracellular electron transfer to HA/MoS₂ coating.” In addition, Umakant Yadav. *et al* claimed that molybdenum disulfide nanosheet reinforced hydroxyapatite nanocomposite scaffolds had been investigated the *in vitro* and *in vivo* osteogenic differentiation, proliferation, and bone regeneration capability (ACS Biomater. Sci. Eng. 2019, 5, 4511–4521). The literature was mainly related with the role of MoS₂/HA composite in cell rather than in bacteria.

Comment 5: How the UV-Vis-NIR samples was prepared? Based on the description, they are not dispersible. Moreover, the MoS₂ or composite of MoS₂ generally exhibit characteristic peaks at ~675 and ~610 nm, 458 and 405 nm related to direct excitonic transition at the K point, which is not observed in the reported spectra.

Reply: Thank you so much for your suggestions.

The UV-Vis-NIR samples were prepared as following steps. First, the samples were fixed at the central of cell. Second, placed the sample cell in the holder that was in the

sample chamber of UV-Vis-NIR spectrophotometer and obtained the spectrum. Third, immediately after use, cleaned the cell with CH₃CH₂OH and returned them to the dessicator.

UV-Vis-NIR analysis of liquid samples and solid samples had great difference. The previous literature was related with MoS₂ film on titanium implant, which also did not exhibit characteristic peaks at ~675 and ~610 nm, 458 and 405 nm (Li, M., Li, L., Su, K., Liu, X. Highly effective and noninvasive near-infrared eradication of a staphylococcus aureus biofilm on implants by a photoresponsive coating within 20 Min. *Advanced Science* **6**, 1900599 (2019)).

Comment 6: In antibacterial activity assay, what quantity of material used, was not mentioned. In other word, what is the MIC/MBC for these nanomaterials? How effective these nanomaterials in compare to other reported systems in compare to both antibacterial activity and osteogenesis?

Reply: Thank you so much for your suggestions.

We performed measurement of minimum inhibitory (MIC) and minimum bactericidal concentration (MBC). The MIC and MBC of HA/MoS₂ was about 5 mg/mL and 10 mg/mL. And we added at **page 8 line 2**: “The antibacterial efficacy of HA/MoS₂ was further performed by minimum inhibitory (MIC) and minimum bactericidal concentration (MBC) (**Supplementary Fig. 10** and **Supplementary Fig. 11**). The MIC and MBC of HA/MoS₂ was about 5 mg/mL and 10 mg/mL.” and **page 25 line 17**: “For measurement of minimum inhibitory concentration (MIC), the MIC values of these nanomaterials were obtained using dilution series of samples. A dilution series of HA/MoS₂ (0 mg/mL, 0.3125 mg/mL, 0.625 mg/mL, 1.25 mg/mL, 2.5 mg/mL, 5 mg/mL, 10 mg/mL, 20 mg/mL, and 40 mg/mL) were prepared, which the steps were identical to the process of synthesis of HA/MoS₂-Ti6. *S. aureus* was diluted to 1.7×10^7 cells mL⁻¹ with LB broth, and 200 μ L of the diluted solution was added into the surface of different samples in the 96-well plate, followed by incubation at 37°C for 6 h. Optical density (OD) was obtained with a microplate reader.

For measurement of minimum bactericidal concentration (MBC), the MBC values

of these nanomaterials were obtained using dilution series of samples. After the same antibacterial process as MIC, bacterial suspension was diluted 60, 000 times, and 20 μL bacterial suspension was coated on LB agar plates and cultured at 37 $^{\circ}\text{C}$ for 24 h. Finally, MBC was obtained from the number of colonies on the plate.”

Supplementary Fig. 10 Sample concentration-OD curves used to determine MIC values. ($n \geq 3$) Source data are provided as a Source Data file.

Method for MIC determination of *S. aureus*. The MIC of HA/MoS₂ was not completely linearly related with the concentration of HA/MoS₂, which was related to the process of synthesis HA/MoS₂-Ti6. The different concentration of precoated HA/MoS₂ coating was prepared, but the effect of laser cladding was not very ideal when the content of HA/MoS₂ was bigger than 10 mg/mL. Those led to weak antibacterial activity of high concentration of HA/MoS₂.

Supplementary Fig. 11 Sample concentration-OD curves used to determine MBC values. ($n \geq 3$) Source data are provided as a Source Data file.

Method for MBC determination of *S. aureus*. The MBC of HA/MoS₂ was not completely linearly related with the concentration of HA/MoS₂, and the reason was same as MIC.

Compared with other reported systems (implant both osteoblastic differentiation of MSCs and biocidal activity of pathogens), our system of HA/MoS₂-Ti6 also had great antibacterial effect and osteoblast differentiation ability. Meanwhile, conventional reported systems mainly focused on the material owing to its inherent antibacterial and osteoblast differentiation poverty. But for our system, the HA/MoS₂-Ti6 had antimicrobial effect through interruption of metabolic pathways and could enhance osteoblast differentiation by regulating the genetic markers associated with osteoblastic differentiation.

Comment 7: Why the HA-MoS₂ composite is more effective than HA or MoS₂ alone?

Reply: Thank you so much for your suggestions. The valence band (VB) maximum and conduction band (CB) minimum for HA/MoS₂ are -4.20 eV and -4.84 eV, respectively. As for MoS₂, the CB and VB for was -4.19 eV and -5.92 eV, respectively. And as for HA, it was not semiconductor, which had no antibacterial ability. And the

bacteria had a biological redox potential (BRP) range of -4.12 to -4.84 eV. When *S. aureus* or *E. coli* contacted with HA/MoS₂, the electrons from bacteria membrane were transferred to HA/MoS₂ due to potential difference, and finally caused bacterial death. HA or MoS₂ alone could not form biohybrid between bacteria and material, and the process of electron transfer would not happen.

Response to Reviewer 3#

Original Comment: In this manuscript, the authors introduce a composite material of hydroxyapatite (HA) and molybdenum sulfide (MoS₂). They hypothesize that this material may be both osteoinductive (promoting differentiation of mesenchymal stem cells into osteoblasts) and antibacterial (through interrupting extracellular aerobic respiration of bacterial pathogens). They set forth to demonstrate this with both in vitro studies (including *Staphylococcus aureus* and *E. coli* as representative gram positive and gram negative pathogens) as well as a rat model of a contaminated tibial defect (*Staphylococcus aureus*).

The authors show in vitro evidence that supports their hypothesis in general (antimicrobial effect through interruption of metabolic pathways, genetic markers associated with osteoblastic differentiation). They show some in vivo evidence of reduced bacterial burden and additional bone growth in an infected defect model although a control arm (surgical defect without infection using the same biomaterials) was not used.

Reply: Thank you so much for your suggestions. Thank you very much for the reviewer's valuable comments and saying that "The authors show in vitro evidence that supports their hypothesis in general (antimicrobial effect through interruption of metabolic pathways, genetic markers associated with osteoblastic differentiation)." The comments are very helpful for us to revise and improve our paper. We have made extensive and careful revisions accordingly, please see the following reply.

Meanwhile, we have already added the *in vivo* experiment of surgical defect without

infection using the same biomaterials as request for more accurate experimental design.

The manuscript has significant grammatical errors.

Reply: Thank you so much for your suggestions. We had already revised our language.

Overall, this is an exciting new direction for the field of device infection. Using metabolism-based therapies rather than traditional antibiotics should circumvent traditional mechanisms of antimicrobial resistance. The in vitro studies performed are thorough; however, I am concerned that the authors have overstated some of their results (for example, expressing reduction of bacteria as a percentage rather than absolute log). In addition, I am concerned that proper controls were not used in the animal model and it is not clear to me if n=6 is properly powered. Overall though, I appreciate the authors' attention to detail and their choice of assays, including genetic analysis.

Reply: Thank you so much for your suggestions. We had revised the expressing reduction bacteria as absolute log, and checked in our manuscript. In addition, we redid animal experiments and n=8.

Specific comments below are as follows:

Comment 1: Page 2, line 28. What does self-powered mean? Please clarify.

Reply: Thank you so much for your suggestions. We were so sorry for our unclear expression of self-powered. Self-powered systems often used to describe systems which did not need an outside energy supply to work (Nature Nanotechnology **2010**, 5(5), 366–373; Advanced Functional Materials **2008**, 18(22), 3553-3567; Nano Today **2010**, 5(6), 512-514). Meanwhile, MoS₂ and HA modified implant took advantage of the energy from metabolic process of *S. aureus* and *E. coli* to active antibacterial process, which did not need extra energy. We thought that the process belonged to self-powered process. We added the explanation about “self-powered” at **page 4 line 21**: “When bacteria were contacted with HA/MoS₂ coating on Ti6, the potential difference between HA/MoS₂-Ti6 and *S. aureus* made the electrons from microbial membranes

transfer to HA/MoS₂ coating surface. The process took advantage of the energy from metabolic process of *S. aureus* and *E. coli* without extra energy to trigger antibacterial activity of HA/MoS₂-Ti6. It was a self-powered anti-infection implant.”

Comment 2: Page 3, line 70. Is molybdenum sulfide truly widely used in biomaterials? I would change this language as this is likely untrue.

Reply: Thank you so much for your suggestions. Molybdenum sulfide could be used in biomaterials due to its unique physical and chemical properties. We also revised our manuscript at **page 3 line 24**: “Moreover, molybdenum sulfide (MoS₂) is a typical layered transition metal dichalcogenide, which is a kind of semiconducting material with unique physical and chemical properties.¹² And those make it applied in biomaterials.¹³”

12 Liu, C. *et al.* Rapid water disinfection using vertically aligned MoS₂ nanofilms and visible light. *Nat. Nanotechnol.* **11**, 1098-1104 (2016).

13 Yadav, V. *et al.* 2D MoS₂ -based nanomaterials for therapeutic, bioimaging, and biosensing applications. *Small* **15**, e1803706 (2019).

Comment 3: Page 4, line 81. This sentence appears incomplete.

Reply: Thank you so much for your suggestions. We had provided detailed information in this sentence at **page 4 line 21**: “When bacteria were contacted with HA/MoS₂ coating on Ti6, the potential difference between HA/MoS₂-Ti6 and *S. aureus* made the electrons from microbial membranes transfer to HA/MoS₂ coating surface. The process took advantage of the energy from metabolic process of *S. aureus* and *E. coli* without extra energy to trigger antibacterial activity of HA/MoS₂-Ti6. It was a self-powered anti-infection implant. Meanwhile, the potential of HA/MoS₂-Ti6 could enhance the viability of MSCs and lead to MSCs differentiate into osteoblast by up-regulating Ca²⁺ level.”

Comment 4: Page 4, line 83. This is likely not the first time a biomaterial has been proposed that is both osteoconductive and antibacterial, although it is still unclear what

"self-powered" implies.

Reply: Thank you so much for your suggestions. We were so sorry for our unclear expression of self-powered. Self-powered systems often used to describe systems which did not need an outside energy supply to work (Nature Nanotechnology **2010**, 5(5), 366–373; Advanced Functional Materials **2008**, 18(22), 3553-3567; Nano Today **2010**, 5(6), 512-514). Meanwhile, MoS₂ and HA modified implant took advantage of the energy from metabolic process of *S. aureus* and *E. coli* to active antibacterial process, which did not need extra energy. We thought that the process belonged to self-powered process. We added the explanation about “self-powered” at **page 4 line 21**: “When bacteria were contacted with HA/MoS₂ coating on Ti6, the potential difference between HA/MoS₂-Ti6 and *S. aureus* made the electrons from microbial membranes transfer to HA/MoS₂ coating surface. The process took advantage of the energy from metabolic process of *S. aureus* and *E. coli* without extra energy to trigger antibacterial activity of HA/MoS₂-Ti6. It was a self-powered anti-infection implant.”

Comment 5: Page 6, line 142. "Coculturing" typically implies culture of two different species or lines; I would suggest that this is culture on a substrate.

Reply: Thank you so much for your suggestions. We had revised in our manuscript at **page 7 line 15**: “culturing bacteria on the substrate” and at **page 15 line 24**: “culturing on the Ti6”.

Comment 6: The phrase "antibacterial efficiency" is used often when describing this material. The mechanism is not clear to me whether the change in bacterial colonies is due to an anti-adhesive effect versus a bacteriostatic or bacteriocidal effect.

Reply: Thank you so much for your suggestions. The mechanism in changing bacterial colonies was mainly due to bacteriocidal effect. And we performed the assays of MIC, MBC, bacterial SEM pictures, water contact angle, and antibacterial activity of different samples to study the mechanism in changing bacterial colonies. We added in our manuscript at **page 9 line 22**: “In addition, the mechanism in the change of bacterial colonies was further performed (**Supplementary Fig. 10-12, Fig. 2a, Fig. 11**). The

results of MIC and MBC indicated that HA/MoS₂-Ti6 had a certain bacteriostatic effect (supplementary Fig. 10-11). Compared with Ti6, the bacteria did not growth rapidly in HA/MoS₂-Ti6. SEM pictures of *S. aureus* after different treatment showed that the cell walls of *S. aureus* were irregular or completely lysed for HA/MoS₂-Ti6 group compared with Ti6 group, suggesting that HA/MoS₂-Ti6 had a certain bactericidal effect. Next, the effect of adhesive poverty of HA/MoS₂-Ti6 on *S. aureus* was further assessed. The HA/MoS₂-Ti6 owned hydrophilic surface due to OH group and roughness, which was beneficial for anti-fouling. HA-Ti6 and MoS₂-Ti6 had similar hydrophilic poverty with HA/MoS₂-Ti6 according to **Supplementary Fig. 5**, but the ability of changing bacterial colonies of HA-Ti6 and MoS₂-Ti6 was poor (**Fig.11**). In additional, the SLM-Ti6 almost had not antibacterial effect, suggesting that the mechanism of the change in bacterial colonies was not due to anti-adhesive effect (**Supplementary Fig. 11**). In a word, the mechanism in bacterial colonies was mainly due to bactericidal effect.”

Supplementary Fig. 10

Supplementary Fig. 11

Supplementary Fig. 5

Supplementary Fig. 12

Fig.11

Fig. 2a

Comment 7: In Figure 1L, it is shown that while statistically significant in effect and a two-log difference in concentration, there still remained between 7-8 log Staph aureus and E coli on the most promising material surface (HA/MoS₂-Ti6). This is far from sterilizing and may clinically still lead to infection. It is misleading to express this difference as a percentage change, I recommend representing it as a log change given bacterial growth kinetics. Please comment.

Reply: Thank you so much for your suggestions. Because of the bacterial concentration we used was far higher than the infection concentration clinically, the remained CFU was not enough. Therefore, we supplied additional experiment about lower infected bacterial concentration to evaluate the antibacterial activity clinically. The results were shown as follows: The c. f. u. of HA/MoS₂-Ti6 group was from 7.32 log to 5.22 log compared with Ti6 group in 10⁵ beginning condition, and from 8.66 log to 6.26 log compared with Ti6 group in 10⁶ beginning condition. We added in our manuscript at **page 7 line 25**: “Lower c. f. u. bacterial was added on the surface of Ti6 and HA/MoS₂-Ti6 to further use to assess the antibacterial ability (**Supplementary Fig. 9**). The c. f. u. of HA/MoS₂-Ti6 group was from 7.32 log to 5.22 log compared with Ti6 group in 10⁵ beginning condition, and from 8.66 log to 6.26 log compared with Ti6 group in 10⁶ beginning condition.”

We revised in our manuscript at **page 7 line 19**: “Compared with Ti6 group, HA–Ti6 group showed about 0.09-log and 0.06-log reduction against *S. aureus* and *E. coli*, respectively. Whereas MoS₂-Ti6 group displayed 0.16-log and 0.2-log reduction against *S. aureus* and *E. coli* due to its layered structure.^{12,28} In contrast, HA/MoS₂–Ti6 group showed 1.81-log reduction of *S. aureus* and 1.88-log reduction of *E. coli* in colony forming unit (c. f. u.), respectively. The methicillin-resistant Staphylococcus aureus (MRSA) treated with HA/MoS₂-Ti6 resulted in 2.36 -log reduction (supplementary Fig. 8).”

12 Liu, C. *et al.* Rapid water disinfection using vertically aligned MoS₂ nanofilms and visible light. *Nat. Nanotechnol.* **11**, 1098-1104 (2016).

28 Karunakaran, S. *et al.* Simultaneous exfoliation and functionalization of 2H-MoS₂ by thiolated surfactants: applications in enhanced antibacterial activity. *J. Am. Chem. Soc.* **140**, 12634-12644 (2018).

Supplementary Fig. 8

Supplementary Fig. 9

Comment 8: The results on page 7 and 8 would be even more exciting if the experiment was replicated with *Staph aureus* mutants/knock outs that lack molecular mechanisms to adapt to anaerobic activity to corroborate the sequencing data.

Reply: Thank you so much for your suggestions. We performed *Staph aureus* knock outs that lack molecular mechanisms to adapt to anaerobic activity, but we did not success to do it. In addition, *pseudomonas aeruginosa* was a strictly aerobic bacterium, which lacked ability to adapt to anaerobic activity.⁴⁷ We did antibacterial assay of Ti6 and HA/MoS₂-Ti6 against *P. aeruginosa* at 6 h and 12 h, suggesting that HA/MoS₂-Ti6 had no ability to clear *P. aeruginosa* infection (**Supplementary Fig. 16**). We added in our manuscript at **page 13 line 23**: “Furthermore, the mechanism of inducing bacteria into anaerobic respiration was evaluated. As shown in **Supplementary Fig. 16**, HA/MoS₂-Ti6 group almost had no antibacterial ability against *pseudomonas aeruginosa* (*P. aeruginosa*) compared with Ti6 group, suggesting that the antibacterial

mechanism of HA/MoS₂-Ti6 group was related to the molecular mechanisms to adapt to anaerobic activity.”

Supplementary Fig. 16

As for the experiment of *Staph aureus* knock outs, we chose *SrrAB* (staphylococcal respiratory response AB) as target gene. *SrrAB* was the key gene to regulation bacterial aerobic and anaerobic respiration. (Tiwaria, N. et al. The *SrrAB* two-component system regulates *Staphylococcus aureus* pathogenicity through redox sensitive cysteines. *Proc. Natl. Acad. Sci.* **117**, 10989-10999, (2020)) The following were the material which was used in this study.

Table 1. Bacterial strains and plasmids used in this study.

Strain or plasmid	Description ^a	Source or reference
Strains		
S. aureus ATCC25923	Staphylococcus aureus	ATCC
S. aureus RN4220	Derived from ATCC25923-4; r ⁻ m ⁺ supE44 ΔlacU169 (Φ80dlacZΔM15) hsdR17	
E. coli DH5a	recA1 endA1 gyrA96 thi-1 relA1	Invitrogen
Plasmids		
pKOR1	Shuttle vector, temperature sensitive, vector for allelic replacement via lambda recombinations and ccdB selection Cm ^R Amp ^R	
pKOR1 Srrab	pKOR1 harboring the Srrab gene, Cm ^R Amp ^R	This study

^a Cm^R, chloramphenicol resistance; Amp^R, ampicillin resistance;

Table. 2 Oligonucleotide Primers for isogenic deletion and complemented mutants construction

No.	Name	Sequence
PL663	srrAB-F-A	GGGGACAAGTTTGTACAAAAAAGCAGGCT CGATTACAAAAACGTATAGCTAATAG
PL664	srrAB-B	ACAGGTCATACCTCCCACACATGCTTTTC
PL665	srrAB-C	GAAAAGCATGTGTGGGAGGTATGACCTGT AATTGAATATAGTTATTTTCAGAACGC
PL666	srrAB-R-D	GGGGACCACTTTGTACAAGAAAGCTGGGT CCTCTTTGATTAAGGGTTGAAAATAC
PL667	SrrAB-KF	AACTGAAGAAGACGATGAAGAAATG
PL668	SrrAB-KD	ATTGTATTATCTTTAGTGTAATTG

And we did several PCR assay.

1. 1 kb upstream and downstream 1 kb fragment PCR: the first round of PCR.

1.2 In the second round of PCR, the PCR product recovered from the above figure was used as a template for PCR.

2. Screening of positive clones

(1) From the DH5a plate transformed with the BP reaction solution, select a single clone, extract the plasmid and directly identify it by electrophoresis.

M: DL2000 DNA marker

1-5: PKOR1*SrrAB* #1-#5 plasmid electrophoresis

6: PKOR1 no load

(2) Selected PKOR1*SrrAB* plasmids for further verification, the primers were PL663/666.

M: DL2000 DNA marker (Takara)

1: PKOR1 no load as template to perform PCR assay

2-6: PKOR1*SrrAB* cloning plasmid as template to perform PCR assay

Chose PKOR1*SrrAB*-#1 to perform sequencing

Sequencing indicated that PKOR1*SrrAB* 1# plasmid was corrected

3. electroporator pKOR1*SrrAB*plasmid into ATCC25923

M 1 2 3 4 5 6

M: DL2000 DNA marker (Takara)

1-6: choosing 的 ATCC25923including PKOR1*SrrAB* plasmid to extract plasmid as template as PCR assay

Primer: PL663/666

Choose #1 to perform next experiment.

4. Identification of gene knockout strains

We did not obtain positive clone until now.

systemic pro-inflammatory immune responses in colonized mice. *Eur. J. Microbiol. Immunol.* **7**, 200-209 (2017).

Comment 9: Page 12, line 291. It is microcomputed tomography, not microtomography.

Reply: Thank you so much for your suggestions. We revised in our manuscript at **page 18 line 25**: "Furthermore, the bone regeneration ability for different samples was assessed via microcomputed tomography (micro-CT) after 4 weeks of implantation.⁴⁶"

⁴⁶ Fu, J. *et al.* Photoelectric-responsive extracellular matrix for bone engineering. *ACS Nano* **13**, 13581-13594 (2019).

Comment 10: Page 12, line 305. Chondrocytes can be precursors to bony development (endochondral ossification) so would be careful about implying that appearance of chondrocytes is not consistent with osteogenesis.

Reply: Thank you so much for your suggestions. We forgot to label the different group. Ti6 group had more chondrocytes than HA/MoS₂-Ti6 group, which was consistent with osteogenesis. And we revised in Fig. 5i-5j. We added at **page 19 line 17**: "As illustrated in Fig. 5i and 5j, the HA/MoS₂-Ti6 with no *S. aureus* group and HA/MoS₂-Ti6 with *S. aureus* group contained many osteoblasts and no chondrocytes in the bone, whereas the Ti6 group with *S. aureus* contained many chondrocytes and the Ti6 group with no *S. aureus* contained some chondrocytes. Compared with Ti6 with *S. aureus* group, the quantitative results indicate that the HA/MoS₂-Ti6 with no *S. aureus* group, HA/MoS₂-Ti6 with *S. aureus* group, and Ti6 with no *S. aureus* group showed 11.13, 9.84, and 6.8, respectively."

Fig. 5i-5j

Comment 11: Page 19, line 478. Please specify if this isolate of *Staphylococcus aureus* is methicillin-susceptible or -resistant.

Reply: Thank you so much for your suggestions. The *Staphylococcus aureus* was methicillin-susceptible *Staphylococcus aureus* ATCC 25923. We specified this part. And we also did the assay of the antibacterial assay of different samples against methicillin-resistant *S. aureus* (MRSA), which the HA/MoS₂-Ti6 also showed great antibacterial efficiency. We did the experiment of HA/MoS₂-Ti6 against MRSA in our manuscript at **page 7 line 23**: “The methicillin-resistant *Staphylococcus aureus* (MRSA) treated with HA/MoS₂-Ti6 resulted in 2.36 -log reduction (**Supplementary Fig. 8**).”

Supplementary Fig. 8

Comment 12: Similar to my above comment on in vitro results, while there was ~two fold log reduction in presence of *Staphylococcus aureus* on the in vitro work, there still remains ~10⁴ bacteria (see Fig 5c). Expressing this reduction as a percentage is misleading given bacterial growth kinetics. While it is reassuring that there was greater bone growth in the treatment arm, it is difficult to know if the remaining 10⁴ bacteria in the long term would compromise the surgical site. In addition, proper matching controls were not performed- there should be two additional groups (Ti6 with no bacterial inoculation and HA/MoS₂-Ti6 with no bacterial inoculation). It is unclear if n=6 per group was correctly powered.

Reply: Thank you so much for your suggestions. Thank you for confirm the results of

antibacterial activity of HA/MoS₂-Ti6. We were sorry that the calculation result of log reduction in presence of staphylococcus aureus was mistaken, and we have revised at **Fig. 5d**. About 10² CFU/mL (far less than 10⁴ CFU/mL) bacteria would not compromise the surgical site (H&E staining and Giemsa staining results could demonstrate this conclusion). We used H&E staining and Giemsa staining of different samples (Ti6 with no *S. aureus* and HA/MoS₂-Ti6 with no *S. aureus*) as control. The result of HA/MoS₂-Ti6 with *S. aureus* was similar with Ti6 with no *S. aureus* group and HA/MoS₂-Ti6 with no *S. aureus* group, suggesting that 10² bacteria would not compromise the surgical site.

Moreover, the antibacterial activity of HA/MoS₂-Ti6 towards less than bacterial concentration was supplied. As shown in **Supplementary Fig. 9**, the c. f. u. of HA/MoS₂-Ti6 group was from 7.32 log to 5.22 log compared with Ti6 group in 10⁵ beginning condition, and from 8.66 log to 6.26 log compared with Ti6 group in 10⁶ beginning condition.

In addition, we redid the animal experiment and added two additional groups (Ti6 with no bacterial inoculation and HA/MoS₂-Ti6 with no bacterial inoculation), and n=8. We added in our manuscript at **page 28 line 20**: “***In vivo* biological evaluation.** Sprague Dawley rats (300–350 g) were bought from the Beijing Huafukang Bioscience Cojnc. Separating the 64 male rats into eight sections (*n* = 8 per section), the eight groups were (1) Ti6 with *S. aureus* after culturing 2 weeks, (2) Ti with no *S. aureus* after culturing 2 weeks, (3) HA/MoS₂-Ti6 with *S. aureus* after culturing 2 weeks, (4) HA/MoS₂-Ti6 with no *S. aureus* after culturing 2 weeks, (5) Ti6 with *S. aureus* after culturing 4 weeks, (6) Ti with no *S. aureus* after culturing 4 weeks, (7) HA/MoS₂-Ti6 with *S. aureus* after culturing 4 weeks, and (8) HA/MoS₂-Ti6 with no *S. aureus* after culturing 4 weeks. As for *S. aureus* infection group, bacterial suspension (20 μL) with a density of 10⁷ CFU mL⁻¹ *S. aureus* was uniformly coated on the surface of Ti6 and HA/MoS₂-Ti6 rods before use. For surgery, pentobarbital sodium salt solution (30 mg/kg, 1% w/w) was used to anesthetize the rats by injection. Then, the different samples were implanted into the tibia near the knee joint. The rats were fed in the same way, and after 2–4 weeks, they were euthanized via an overdose of chloral hydrate.

Spread plate analysis and histological analysis. To determine the antibacterial efficiency of Ti6 and HA/MoS₂-Ti6 rods, the rods (Ti with *S. aureus* and HA/MoS₂-Ti6 with *S. aureus*) were removed after 2 weeks, rolled on a standard agar plate for four rounds, and then cultured for 24 h at 37°C. The rods after rolling on a standard plate were plated in small glass vials, and then cultured for 24 h at 37°C. The bacterial colonies and glass vials were photographed on a rolling trace using a digital camera. Meanwhile, H&E and Giemsa staining were used to determine the bacterial contamination of bone-tissue and bone implants after 2 weeks. The samples without bacteria of H&E and Giemsa staining were used to be control group, which was used to assess the influence of bacteria on bone tissue. A fluorescence microscope was used to analyze the histopathological microtomography.”

And at **page 18 line 10**: “The inflammatory response and remaining bacteria in the bone tissue around the implant were evaluated via hematoxylin and eosin (H&E) and Giemsa staining after 14 days of implantation (**Fig. 5a** and **Fig. 5b**).⁴⁵ H&E staining was used to observe neutrophils, and the tissue around Ti6 with *S. aureus* had many neutrophils (Fig. 5a, blue arrow). On the contrary, the number of neutrophils in HA/MoS₂-Ti6 with *S. aureus* group was less than that in Ti6 with *S. aureus* group because of the effective antibacterial efficacy of composite coating. The Ti6 with no *S. aureus* and HA/MoS₂-Ti6 with no *S. aureus* had fewer neutrophils. Giemsa staining (**Fig. 5b**, black arrow) was used to observe the number of bacteria around the tissue of the implant. The Ti6 with no *S. aureus* group and HA/MoS₂-Ti6 with no *S. aureus* group had no bacteria. Compared to Ti6 with *S. aureus* group, few bacteria in HA/MoS₂-Ti6 with *S. aureus* group were observed, further proving the highly effective antibacterial ability of HA/MoS₂-Ti6 group *in vivo*. The results of H&E and Giemsa staining indicated that the remaining bacteria in long term would not compromise the surgical site. The bacterial spread plate and colonies were further used to evaluate the antibacterial ability of different samples *in vivo* (**Fig. 5c** and **Fig. 5d**). Compared to the Ti6 rod with *S. aureus*, the bacteria in HA/MoS₂-Ti6 with *S. aureus* group would be 2.65-log reduction. Furthermore, the bone regeneration ability for different samples was assessed *via*

microcomputed tomography (micro-CT) after 4 weeks of implantation.⁴⁶ Three cylindrical regions around the implant—with a diameter of 2.51 mm and a thickness of 0.40 mm—were selected, and three- and two-dimensional images were reconstructed with a special software program to reduce errors (**Fig. 5e**). The newly formed bone tissues (Obj. V [object volume]/TV [tissue volume]) were used to analyze the bone mass. HA/MoS₂-Ti6 with *S. aureus* group (44.62%) and HA/MoS₂-Ti6 with no *S. aureus* group (51.67%) showed significantly more newly formed bone tissues than did Ti6 with *S. aureus* group (25.40%) and Ti6 with no *S. aureus* group (35.13%), as shown in **Fig. 5f**. Methylene blue-acid magenta staining and Safranin-O/Fast Green staining were used to assess the histopathological conditions around the implant after 4 weeks (**Fig. 5g–5h**). As for Methylene blue-acid magenta staining, the red color represented the mineralized bone tissue in methylene blue-acid fuchsin staining. Compared to Ti6 with *S. aureus* group, HA/MoS₂-Ti6 with *S. aureus* group, HA/MoS₂-Ti6 with no *S. aureus* group, and Ti6 with no *S. aureus* group showed better bone-implant contact. The quantitative results showed that the HA/MoS₂-Ti6 with *S. aureus* group (89.57%) and HA/MoS₂-Ti6 with no *S. aureus* group (85.50%) had excellent bone contact, which was much higher than the corresponding value of the Ti6 with no *S. aureus* group (44.18%) and Ti6 with *S. aureus* group (12.12%). As regards Safranin-O/Fast Green staining, osteogenesis was represented by green and cartilage by red or orange. As illustrated in **Fig. 5i** and **Fig. 5j**, the HA/MoS₂-Ti6 with no *S. aureus* group and HA/MoS₂-Ti6 with *S. aureus* group contained many osteoblasts and no chondrocytes in the bone, whereas the Ti6 group with *S. aureus* contained many chondrocytes and the Ti6 group with no *S. aureus* contained some chondrocytes. Compared with Ti6 with *S. aureus* group, the quantitative results indicate that the HA/MoS₂-Ti6 with no *S. aureus* group, HA/MoS₂-Ti6 with *S. aureus* group, and Ti6 with no *S. aureus* group showed 11.13, 9.84, and 6.8, respectively. The *in vivo* toxicology of major organs—including the heart, liver, spleen, lung, and kidney—was detected by H&E staining after 4 weeks. As shown in **Fig. 5k**, no signs of organ damage were found, indicating that there was no obvious histological toxicology. These results show that the composite coating of HA/MoS₂ group not only had the self-antibacterial performance of inhibiting

the growth of pathogens but also had good osteoinductivity, which simultaneously made HA/MoS₂ group a promising surface system to solve the two major concerns of bacterial infection and osseointegration.”

- 45 Li, Y. *et al.* Eradicating multidrug-resistant bacteria rapidly using a multi functional g-C₃N₄@ Bi₂S₃ nanorod heterojunction with or without antibiotics. *Adv. Funct. Mater.* **29**, 1900946 (2019).
- 46 Fu, J. *et al.* Photoelectric-responsive extracellular matrix for bone engineering. *ACS Nano* **13**, 13581-13594 (2019).

Figure 5

Lastly, we would like to thank the Editor and all the Reviewers again for their time and effort in helping us improve the quality of this manuscript. It is greatly appreciated. We hope that our responses are satisfactory and the revised manuscript could meet the standard of *Nature Communications*.

Thank you very much.

Yours sincerely,

Shuilin

Shuilin Wu, Professor

School of Materials Science & Engineering

Tianjin University

Tianjin

** See Nature Research's author and referees' website at www.nature.com/authors for information about policies, services and author benefits.

REVIEWER COMMENTS

Reviewer #3 (Remarks to the Author):

In general, this manuscript has been strengthened by the efforts of its authors. In particular, the study design of their in vitro work has been much improved by including important control groups. By including these groups, they have generated further support to their hypothesis that their biomaterial may have both osteoinductive and antimicrobial properties. However, this new in vivo data is confusing in that it does not appear all groups were statistically compared to one another. It would be further strengthened by clarifying the statistical relationships between the 4 groups.

Ultimately, the grammatical errors throughout the text make interpreting sections challenging. For example, in the abstract alone, note grammatical errors in lines 27-28, 34, and 39. Errors are rife throughout the manuscript and interfere with understanding the authors' interpretation of their data.

While I believe I understand the authors' explanation behind their terminology of "self-powered," I still find it non-intuitive to describe their phenomenon and it appears the other reviewers agree that it is a confusing term. I would consider another phrase, such as "self-activating."

Specific comments as follows:

Page 3 Line 77 and Page 4 Line 88: I do not understand the sentence "And those make it applied in biomaterials."

Page 4 Last Paragraph: Given the inconsistent tenses, this paragraph was difficult to read.

Page 10 Line 225: I am not familiar with the term "adhesion poverty" or several lines down "hydrophilic poverty."

Page 10 Line 228: "...but the ability of changing bacterial colonies of HA-Ti6 and MoS2-Ti6 was poor (Fig.11)." What does this mean? What is the ability of changing bacterial colonies? This paragraph in general is difficult to read; it is hard to understand what conclusions the authors are trying to reach and how they have reached them given the text.

Page 18 Line 418: The fact that there are less neutrophils present on H&E does not necessarily support the claim that "the remaining bacteria in long term would not compromise the surgical site." For example, less neutrophils recruited to the site may mean that the remaining bacteria have been able to more efficiently create a biofilm matrix or enter a quiescent state such that this will become the site of a chronic infection. The histology does not necessarily support the claim of the authors.

Figure 5F. Are the groups statistically significantly different from one another? My understanding is that the astrix indicate that the other three groups have significantly greater % than the infected Ti6 group. This also applies to Figures 5h and j; it is not clear which groups are being compared for statistical significance.

Response to Reviewer 3#

In general, this manuscript has been strengthened by the efforts of its authors. In particular, the study design of their in vitro work has been much improved by including important control groups. By including these groups, they have generated further support to their hypothesis that their biomaterial may have both osteoinductive and antimicrobial properties. However, this new in vivo data is confusing in that it does not appear all groups were statistically compared to one another. It would be further strengthened by clarifying the statistical relationships between the 4 groups.

Reply: Thank you so much for positive comment about "In general, this manuscript has been strengthened by the efforts of its authors. In particular, the study design of their in vitro work has been much improved by including important control groups. By including these groups, they have generated further support to their hypothesis that their biomaterial may have both osteoinductive and antimicrobial properties. "

The reviewer also said that "this new in vivo data is confusing in that it does not appear all groups were statistically compared to one another. It would be further strengthened by clarifying the statistical relationships between the 4 groups." As request, we have clarified the statistical relationships between the 4 groups as follows:

Figure 6 e The 2D and 3D pictures of HA/MoS₂-Ti6 and Ti6 groups measured by Micro-CT. **f** Bone volume (BV)/tissue volume (TV) values of HA/MoS₂-Ti6 and Ti6 groups. **g** Methylene blue-acid magenta staining of the newly formed bone tissues on

the bone-implant interface, scale bar = 100 μm . **h** Bone area ratios of samples calculated from the methylene blue-acid magenta staining. **i** Safranin-O/Fast Green staining. The green color is osteogenesis, and the red or orange color is cartilage. **j** Histomorphometric measurements of osteogenesis. The area was taken from 20 μm around the implant.

Ultimately, the grammatical errors throughout the text make interpreting sections challenging. For example, in the abstract alone, note grammatical errors in lines 27-28, 34, and 39. Errors are rife throughout the manuscript and interfere with understanding the authors' interpretation of their data.

Reply: Thank you so much for your comments. We have checked our manuscript carefully and revised the grammar mistakes throughout the manuscript as follows:

1. **Page 2 Line 26:** the original sentence: "Current strategies are often hard to satisfy osseointegration and antibacterial requirements simultaneously." Changing into "Clinically, it is difficult to endow implants with excellent osteogenic ability and antibacterial activity simultaneously."

2. **Page 2 Line 29:** the original sentence: "RNA sequencing reveals that the transcriptional regulations of *S. aureus* express less aerobic respiration–related genes and more anaerobic respiration–related genes when interfaces with HA/MoS₂ coating." Changing into "The electron transfer between the bacteria and HA/MoS₂ coating is triggered when bacteria contacted with materials. And the RNA sequencing data further reveal that the expression level of anaerobic respiration–related genes is up-regulated and the expression level of aerobic respiration–related genes is down-regulated when bacteria adhere to the surface of implants."

3. **Page 2 Line 39:** the original sentence: "HA/MoS₂ modification can be a novel and viable method to design implants with the ability of antibacterial and osseointegration." Changing into "Surface modification by HA/MoS₂ coating will be a promising method to endow metallic implants with self-activating anti-infection performance and osteogenic ability simultaneously."

4. **Page 4 Line 81:** the original sentence: "2D MoS₂ nanostructures had a tunable electronic energy state." Changing into "2D MoS₂ can have a tunable electronic energy state due to its tunable nanosheet-like structure."

5. **Page 4 Line 82:** the original sentence: "nanostructured MoS₂ would benefit the separation of electron-hole pairs by decreasing the distances for electrons and holes to diffuse to the surface of the materials and also increasing the reaction sites." Changing into "nanostructured MoS₂ can benefit the separation of electron-hole pairs by not only decreasing the distances of electrons and holes to move to the surface of the materials but also increasing the active sites."

6. **Page 4 Line 84:** the original sentence: "Furthermore, Mo is the basic trace element of many enzymes, and S is a common biological element in cells, which makes MoS₂ be an excellent candidate in biomaterials." Changing into "It is well known that Mo is the basic trace element of many enzymes, and S is a common biological element in cells, which make MoS₂ be potentially attractive for biomedical applications."

7. **Page 5 Line 105:** the original sentence: "Meanwhile, the potential of HA/MoS₂-Ti6 could enhance the viability of MSCs and lead to MSCs differentiate into osteoblast by up-regulating Ca²⁺ level." Changing into "Meanwhile, the HA/MoS₂-Ti6 could enhance the viability of MSCs and promote the osteogenic differentiation of MSCs by up-regulating Ca²⁺ level."

8. **Page 5 Line 114:** the original sentence: "Scanning electron microscopy (SEM) was obtained to demonstrate that HA/MoS₂ coating completely covered Ti6." Changing into "The scanning electron microscopy (SEM) image was obtained to demonstrate that Ti6 substrate was completely covered by HA/MoS₂ coating."

9. **Page 5 Line 124:** the original sentence: "there were obvious Ca, P peaks, and Mo, S peaks in the HA-Ti6 and MoS₂-Ti6 spectra, respectively." Changing into "there were obvious peaks of Ca and P in the HA-Ti6 spectrum and peaks of Mo and S in the MoS₂-Ti6 spectrum."

10. **Page 5 Line 127:** the original sentence: "The 229.2 and 232.4 eV peaks corresponded to Mo 3d_{5/2} and Mo 3d_{3/2} severally, whereas the 162.0 and 163.2 eV peaks corresponded to S 2p_{3/2} and S 2p_{1/2}, which matched MoS₂, manifesting the existence of

MoS₂ in the coating." Changing into "The peaks at 229.2 eV and 232.4 eV corresponded to Mo 3d_{5/2} and Mo 3d_{3/2} orbitals severally, whereas the peaks at 162.0 eV and 163.2 eV corresponded to S 2p_{3/2} and S 2p_{1/2} orbitals, which belonged to MoS₂. The result demonstrated the existence of MoS₂ in the coating."

11. **Page 6 Line 135:** the original sentence: "HA/MoS₂-Ti6 had no signal of vacancies compared with Ti6, suggesting that the structure of MoS₂ did not change in this experiment." Changing into "There were no signals of vacancies for HA/MoS₂-Ti6, suggesting that the vacancies were not formed after the treatment of laser cladding and CVD."

12. **Page 6 Line 138:** the original sentence: "The water contact angle was measured to evaluate the hydrophilicity of the surfaces." Changing into "The water contact angle was measured to evaluate the surface hydrophilicity or hydrophobicity."

13. **Page 7 Line 176:** the original sentence: "Whereas MoS₂-Ti6 group displayed 0.16-log and 0.2-log reduction against *S. aureus* and *E. coli* due to its layered structure." Changing into " MoS₂-Ti6 group displayed 0.16-log and 0.2-log reduction in bacterial counts against *S. aureus* and *E. coli*, which was possibly ascribed to the layered structure of MoS₂."

14. **Page 7 Line 178:** the original sentence: "In contrast, HA/MoS₂-Ti6 group showed 1.81-log reduction of *S. aureus* and 1.88-log reduction of *E. coli* in colony forming unit (c. f. u.), respectively." Changing into "In contrast, there were only a few bacterial colonies in HA/MoS₂-Ti group, which showed 1.81-log and 1.88-log reduction in bacterial counts against *S. aureus* and *E. coli*, respectively, suggesting its great broad-spectrum antibacterial performance."

15. **Page 8 Line 180:** the original sentence: "The methicillin-resistant *Staphylococcus aureus* (MRSA) treated with HA/MoS₂-Ti6 resulted in 2.36-log reduction (Supplementary Fig. 8)." Changing into "Besides, the methicillin-resistant *Staphylococcus aureus* (MRSA) treated with HA/MoS₂-Ti6 showed 2.36-log reduction in bacterial counts (Supplementary Fig. 8)."

16. **Page 8 Line 185:** the original sentence: "The c. f. u. of HA/MoS₂-Ti6 group was from 7.32 log to 5.22 log compared with Ti6 group in 10⁵ beginning condition, and

from 8.66 log to 6.26 log compared with Ti6 group in 10^6 beginning condition." Changing to "The bacterial counts in log (CFU mL⁻¹) in HA/MoS₂-Ti6 group were reduced from 7.32-log to 5.22-log after 6 h culturing with an initial bacterial concentration of 10^5 CFU mL⁻¹, and from 8.66-log to 6.26-log after 6 h cultivation with an initial concentration of 10^6 CFU mL⁻¹."

17. **Page 10 Line 214:** the original sentence: "As shown in **Fig. 2a**, compared to Ti6, whether it was *S. aureus* or *E. coli*, more green spots could be seen from HA-Ti6 group and MoS₂-Ti6 group, suggesting more living bacteria and a small number of dead bacteria." Changing into "As shown in **Fig. 3a**, compared to Ti6 group, there were many live bacteria and little dead bacteria in the two groups of HA-Ti6 and MoS₂-Ti6."

18. **Page 10 Line 216:** the original sentence: "In contrast, HA/MoS₂-Ti6 showed a very small number of green spots, suggesting the best antibacterial ability." Changing into "In contrast, HA/MoS₂-Ti6 group showed few live bacteria, suggesting its great antibacterial ability."

19. **Page 11 Line 235:** the original sentence: "Next, the effect of adhesive ability of HA/MoS₂-Ti6 on *S. aureus* was further assessed. The HA/MoS₂-Ti6 owned hydrophilic surface due to OH group and roughness, which was beneficial for anti-fouling." Changing into "Next, the influence of antifouling ability on antibacterial activity was evaluated. Generally, the good hydrophilicity is beneficial for developing an antifouling surface. In this work, as shown in **Supplementary Fig. 5**, the HA/MoS₂-Ti6 owned a good hydrophilic surface because of many hydroxyl groups on the surface, and similarly, both HA-Ti6 and MoS₂-Ti6 had good hydrophilicity."

20. **Page 12 Line 263:** the original sentence: "This suggests that the bacterial redox equilibrium was disturbed when bacteria were in contact with HA/MoS₂-Ti6 group." Changing into "Those results showed that bacteria could effectively produce intracellular ROS when they adhered to the surface of HA/MoS₂-Ti6."

21. **Page 14 Line 303:** the original sentence: "As shown in **Supplementary Fig. 15**, when bacteria were cultured on the surface of HA/MoS₂-Ti6, the zeta potential would turn more negative compared with bacteria cultured on the surface of Ti6." Changing

into "As shown in **Supplementary Fig. 15**, when bacteria were cultured on the surface of HA/MoS₂-Ti6, the zeta potential **turned** more negative compared with bacteria cultured on the surface of Ti6."

22. Page 14 Line 309: the original sentence: "After contacted with HA/MoS₂-Ti6 group, the ATP of bacteria decreased the most, which further indicated that HA/MoS₂-Ti6 group effectively destroyed the normal respiration of bacteria." **Changing into** "In this work, after bacteria adhered to the surface of HA/MoS₂-Ti6, the bacteria in this group exhibited the lowest ATP level among all groups, which further indicated that HA/MoS₂-Ti6 effectively destroyed the normal respiration of bacteria."

23. Page 14 Line 319: the original sentence: "The remaining electrons were extracted *via* HA/MoS₂-Ti6 group because of potential differences through membrane cytochromes or electrically conductive pili." **Changing into** "The remaining electrons were transferred to HA/MoS₂-Ti6 through membrane cytochromes or electrically conductive pili because of the potential difference between bacteria and HA/MoS₂-Ti6."

24. Page 17 Line 377: the original sentence: "These results indicate that HA-Ti6 group, MoS₂-Ti6 group, and HA/MoS₂-Ti6 group had great biocompatibility." **Changing into** "These results suggested that the cell proliferation was mainly due to the enhanced mitochondrial metabolism rather than three-dimensional structure caused by laser cladding."

25. Page 17 Line 383: the original sentence: "This was related to the osteoconductivity of HA and its potential, which activated the Wnt/ β -catenin and Wnt/Ca²⁺ pathways." **Changing into** "This was related to the osteoconductive HA and the potential of HA/MoS₂-Ti6, which activated the Wnt/ β -catenin and Wnt/Ca²⁺ pathways."

26. Page 19 Line 428: the original sentence: "The bacterial spread plate and colonies were further used to evaluate the antibacterial ability of different samples *in vivo* (**Fig. 5c** and **Fig. 5d**)."**Changing into** " The bacterial spread plate and colonies were further used to **quantify** the antibacterial ability of different samples *in vivo* (**Fig. 6c** and **Fig. 6d**)."

27. Page 20 Line 430: the original sentence: "Compared to the Ti6 rod with *S. aureus*,

the bacteria in HA/MoS₂-Ti6 with *S. aureus* group would be 2.65-log reduction." Changing into "Compared to the group of Ti6+*S. aureus*, the bacteria in HA/MoS₂-Ti6 + *S. aureus* group were 2.65-log reduction in bacterial counts."

28. **Page 23 Line 482:** the original sentence: "In this study, the integration of semiconductor coating with bacterial metabolism and cell membranes provided a new angle for the study of bacterial growth and cell fate." Changing into " In summary, the synthesized HA/MoS₂ coating by laser cladding took advantage of the difference of energy metabolism between MSCs and bacteria to kill pathogenic bacteria and promote the bone regeneration on Ti implants simultaneously."

29. **Page 23 Line 486:** the original sentence: "Differential gene expression analysis revealed that *S. aureus* altered gene expression in response to the changes of electron receptors, in which *S. aureus* modulated the metabolism pathway from aerobic to anaerobic respiration." Changing into "Differential gene expression analysis revealed that after bacteria adhered to HA/MoS₂-Ti6, the coating of HA/MoS₂ not only altered the gene expression of bacteria in response to the changes of electron receptors, but also modulated their metabolism pathway from aerobic to anaerobic respiration."

While I believe I understand the authors' explanation behind their terminology of "self-powered," I still find it non-intuitive to describe their phenomenon and it appears the other reviewers agree that it is a confusing term. I would consider another phrase, such as "self-activating."

Reply: Thank you so much for your suggestion. We have revised our title at **Page 1 Line 1:** "Self-activating anti-infection implant", at **Page 2 Line 27:** "the self-activating implants modified with hydroxyapatite (HA)/MoS₂ coating", and at **Page 5 Line 106:** "The discovery of self-activating anti-infection implants is expected to spur the development of orthopedic implants in clinics."

Specific comments as follows:

Page 3 Line 77 and Page 4 Line 88: I do not understand the sentence “And those make it applied in biomaterials.”

Reply: Thank you so much for your comment. We were so sorry for our unclear expression. Besides its excellent biocompatibility, molybdenum sulfide (MoS₂) can exhibit metallic or semiconducting properties by tuning its structure, which makes it be widely used in biomedical filed. We have revised it at **Page 3 Line 75**: "It has been reported that besides its excellent biocompatibility, molybdenum sulfide (MoS₂) can exhibit metallic or semiconducting properties by tuning its structure, which makes it be used in biomedical filed including anti-cancer therapy, antibacterial therapy and diagnostics.¹²⁻¹⁶ Additionally, the unique physical and photo-electrical properties can endow this material with excellent bactericidal ability against both Gram-positive and Gram-negative bacteria.^{13,17,18"}

12 Yadav, V. *et al.* 2D MoS₂ -based nanomaterials for therapeutic, bioimaging, and biosensing applications. *Small* **15**, e1803706 (2019).

13 Karunakaran, S. *et al.* Simultaneous exfoliation and functionalization of 2H-MoS₂ by thiolated surfactants: applications in enhanced antibacterial activity. *J. Am. Chem. Soc.* **140**, 12634-12644 (2018).

14 Pandit, S. *et al.* High antibacterial activity of functionalized chemically exfoliated MoS₂. *ACS Appl. Mater. Interfaces* **8**, 31567-31573 (2016).

15 Yang, X. *et al.* Antibacterial activity of two-dimensional MoS₂ sheets. *Nanoscale* **6**, 10126-10133 (2014).

16 Bazaka, K. *et al.* MoS₂-based nanostructures: synthesis and applications in medicine. *J. Phys. D: Appl. Phys.* **52**, 183001 (2019).

17 Liu, C. *et al.* Rapid water disinfection using vertically aligned MoS₂ nanofilms and visible light. *Nat. Nanotechnol.* **11**, 1098-1104 (2016).

18 Kong, X. *et al.* Graphitic carbon nitride-based materials for photocatalytic antibacterial application. *Mater. Sci. Eng., R* **145**, 100610 (2021).

Page 4 Last Paragraph: Given the inconsistent tenses, this paragraph was difficult to read.

Reply: Thank you so much for your comment. We have revised it in our manuscript at **Page 4 Line 95**: "In this study, we hypothesized that the HA/MoS₂ coating could endow the metallic implants with excellent osteogenic ability and antibacterial activity simultaneously. The Ti6Al4V (Ti6) implant modified with HA/MoS₂ coating was prepared by laser cladding and chemical vapor deposition (CVD). In addition, the

HA/MoS₂ coating could effectively eradicate bacteria by extracting electrons from bacteria and simultaneously accelerate bone regeneration by promoting the osteogenic differentiation of MSCs (**Fig. 1**). When bacteria adhered to HA/MoS₂ coating on Ti6, the potential difference between HA/MoS₂-Ti6 and bacteria made the electrons from bacterial membranes transfer to the surface of HA/MoS₂ coating. The process took advantage of the energy from metabolic process of *S. aureus* and *E. coli* without extra energy to trigger antibacterial activity of HA/MoS₂-Ti6. It was a self-activating process of anti-infection. Meanwhile, the HA/MoS₂-Ti6 could enhance the viability of MSCs and promote the osteogenic differentiation of MSCs by up-regulating Ca²⁺ level. The discovery of self-activating anti-infection implants is expected to spur the development of orthopedic implants in clinics."

Page 10 Line 225: I am not familiar with the term “adhesion poverty” or several lines down “hydrophilic poverty.”

Reply: Thank you so much for your suggestions. We were sorry for our wrong description about these two terms. We have revised "adhesion poverty" into "antifouling ability" at **Page 11 Line 235**. Generally, the good hydrophilicity is beneficial for developing an antifouling surface. Anti-fouling ability of material means the effect of inhibition of pathogen to adhere on the surface of substrate.

And we have also revised "hydrophilic poverty" into "hydrophilicity" at **Page 11 Line 236**. hydrophilicity was a useful material property, and hydrophilic surface could attract water.

Page 10 Line 228: “...but the ability of changing bacterial colonies of HA-Ti6 and MoS₂-Ti6 was poor (Fig.11).” What does this mean? What is the ability of changing bacterial colonies? This paragraph in general is difficult to read; it is hard to understand what conclusions the authors are trying to reach and how they have reached them given the text.

Reply: Thank you so much for your professional suggestions. In this sentence, "the

ability of changing bacterial colonies of HA-Ti6 and MoS₂-Ti6 was poor" means that the antibacterial ability of HA-Ti6 and MoS₂-Ti6 was weak. The ability of changing bacterial colonies meant the antibacterial ability. And we also revised it in our manuscript at **Page 11 Line 239**: "However, the former exhibited highly effective antibacterial efficacy while the latter two had no antibacterial effect (**Fig. 2I**)."

The final conclusion was that antibacterial activity predominantly resulted from the bactericidal ability of HA/MoS₂-Ti6. And we analyzed the influence of bacteriostatic effect, bactericidal ability, antifouling effect, and the structure produced by laser cladding on antibacterial activity of HA/MoS₂-Ti6, respectively.

Meanwhile, we have also revised this paragraph for more easily to read at **Page 10 Line 226**: "In addition, the antibacterial mechanism of HA/MoS₂-Ti6 against *S. aureus* was further analyzed in detail (**Supplementary Fig. 5, Supplementary Fig. 10-12, Fig. 3a, Fig. 2I**). First, the values of MIC and MBC of HA/MoS₂-Ti6 were about 5 mg mL⁻¹ and 10 mg mL⁻¹, respectively. It is known that the antibacterial agents are considered bactericidal if the value of MBC is < 4 times the value of MIC.¹³ The ratio of MBC to MIC was 2, indicating that HA/MoS₂-Ti6 had a certain bacteriostatic effect (**Supplementary Fig. 10-11**). Second, the influence of HA/MoS₂-Ti6 on bacterial morphology was further assessed by SEM. The bacterial membranes were irregular or completely broken after culturing on the surface of HA/MoS₂-Ti6 for 6 h, suggesting the bactericidal ability of HA/MoS₂-Ti6 (**Fig. 3a**). Next, the influence of antifouling ability on antibacterial activity was evaluated. Generally, the good hydrophilicity is beneficial for developing an antifouling surface. In this work, as shown in **Supplementary Fig. 5**, the HA/MoS₂-Ti6 owned a good hydrophilic surface because of many hydroxyl groups on the surface, and similarly, both HA-Ti6 and MoS₂-Ti6 had good hydrophilicity. However, the former exhibited highly effective antibacterial efficacy while the latter two had no antibacterial effect (**Fig. 2I**). These results suggested that the antibacterial ability of HA/MoS₂-Ti6 was not caused by the antifouling effect. Additionally, the laser cladding induced structure on Ti6 (SLM-Ti6) almost had no antibacterial effect (**Supplementary Fig. 12**), suggesting the antibacterial ability of HA/MoS₂-Ti6 was not caused by the laser induced structure. All in all, the antibacterial

activity predominantly resulted from the bactericidal ability of HA/MoS₂-Ti6."

Supplementary Fig. 10. OD₆₀₀ of *S. aureus* co-cultured with HA/MoS₂ under different preparation conditions after culturing for 6 h. (n≥3) Source data are provided as a Source Data file.

Supplementary Fig. 11 Sample concentration- Log₁₀ (CFU mL⁻¹) curves used to determine MBC values. (n≥3) Source data are provided as a Source Data file.

Supplementary Fig. 5 Water contact angle of Ti6, HA-Ti6, MoS₂-Ti6 and HA/MoS₂-Ti6. (n = 3, mean ± s. d. **p < 0.01). Source data are provided as a Source Data file.

Supplementary Fig. 12 The antibacterial activity of Ti6, and SLM-Ti6 against *S. aureus*. Data represent means ± s. d. t test. (n≥3). Source data are provided as a Source Data file.

Fig. 2 The antibacterial results of Ti6, HA-Ti6, MoS₂-Ti6 and HA/MoS₂-Ti6 against *S. aureus* and *E. coli*.

Fig. 3a Fluorescent images of stained bacteria (scale bar = 50 μm) and FE-SEM morphologies (scale bar = 1 μm) after treatment on various surface for *S. aureus* and *E. coli*.

Page 18 Line 418: The fact that there are less neutrophils present on H&E does not necessarily support the claim that “the remaining bacteria in long term would not compromise the surgical site.” For example, less neutrophils recruited to the site may mean that the remaining bacteria have been able to more efficiently create a biofilm matrix or enter a quiescent state such that this will become the site of a chronic infection. The histology does not necessarily support the claim of the authors.

Reply: Thank you so much for your suggestions. Less neutrophils were recruited to the tissue surrounding of HA/MoS₂-Ti6, which meant weak inflammatory response. It might be beneficial to form biofilm. However, HA/MoS₂-Ti6 had antibacterial ability, which could limit the remaining bacteria to form the biofilm. Meanwhile, we have deleted original statement and revised it at **Page 19 Line 424**: "As a result, the initially added *S. aureus* caused a much weaker inflammation in the implant of HA/MoS₂-Ti6 but a strong inflammation reaction in Ti6 group. The Giemsa staining further indicated that HA/MoS₂-Ti6 killed the most bacteria with few bacteria remaining. Consequently, it was hard to form the biofilm on the surface of HA/MoS₂-Ti6."

Figure 5F. Are the groups statistically significantly different from one another? My understanding is that the astrix indicate that the other three groups have significantly greater % than the infected Ti6 group. This also applies to Figures 5h and j; it is not clear which groups are being compared for statistical significance.

Reply: Thank you so much for your comment. The results indicated that the groups statistically were significantly different from one another in **Fig. 6f** (originally **Fig. 5f**). Based on your request, we have clarified the statistical relationships between the 4 groups in **Fig. 6f**, **Fig. 6h** and **Fig. 6j**.

Figure 6 **e** The 2D and 3D pictures of HA/MoS₂-Ti6 and Ti6 groups measured by Micro-CT. **f** Bone volume (BV)/tissue volume (TV) values of HA/MoS₂-Ti6 and Ti6 groups. **g** Methylene blue-acid magenta staining of the newly formed bone tissues on the bone-implant interface, scale bar = 100 μm. **h** Bone area ratios of samples calculated from the methylene blue-acid magenta staining. **i** Safranin-O/Fast Green staining. The green color is osteogenesis, and the red or orange color is cartilage. **j** Histomorphometric measurements of osteogenesis. The area was taken from 20 μm around the implant.

REVIEWERS' COMMENTS

Reviewer #3 (Remarks to the Author):

The results of this study are noteworthy and significant for those that study device-related infection.

The work is difficult to compare to established literature- the MIC and MBC values reported were on the scale of mg/mL. For traditional antibiotics, the concentrations required to achieve MIC are typically on the scale of mg/L- otherwise, clinically levels cannot be achieved without significant host toxicity. The authors should comment on why they have found an in vivo effect of their material even though the in vitro concentration of effectiveness requires concentration thousands-fold higher than that of typical antimicrobial agents.

The authors have clarified some of their other data such as statistical significance in their in vivo studies which is helpful.

Response to Reviewer #3

The results of this study are noteworthy and significant for those that study device-related infection.

Reply: Thank you very much for your valuable and professional comments on our work. Thanks so much for the positive evaluation.

The work is difficult to compare to established literature- the MIC and MBC values reported were on the scale of mg/mL. For traditional antibiotics, the concentrations required to achieve MIC are typically on the scale of mg/L- otherwise, clinically levels cannot be achieved without significant host toxicity. The authors should comment on why they have found an in vivo effect of their material even though the in vitro concentration of effectiveness requires concentration thousands-fold higher than that of typical antimicrobial agents.

Reply: Thank you very much for your professional suggestion. The MIC and MBC values of HA/MoS₂-Ti6 were thousands of times higher than traditional antibiotics. On the one hand, HA/MoS₂-Ti6 took advantage of film against *S. aureus*, not power. On the other hand, coating against bacteria was mainly involved with material quality, according to previous literature (*Nat. Commun.* **9**, 1-12 (2018); *Angew. Chem., Int. Ed.* **132**, 6051-6055 (2020)). In detail, traditional antibiotics interacted with bacteria in solution, in which the antibiotics could fully contact bacteria. But as for HA/MoS₂-Ti6, the contact area between the material in the coating and the bacteria is tiny, and a certain coating thickness is usually required in preparation to ensure subsequent stability and longevity. Second, the coating concentration referred to the concentration of the precursor solution used in preparation. More importantly, the quality of material on the surface of the titanium implant was related to concentration and volume. We reduced the precursor concentration but increasing the volume of the material, the same quality of the material on the coating surface could also be obtained. It was not accurate to use the concentration alone to describe the material of the coating surface and compare the antibacterial ability with traditional antibiotics.

In addition, we assessed the material biosafety in vitro and in vivo. HA/MoS₂-Ti6 had great antibacterial ability and biocompatibility. This material had no apparent histological toxicology and great potential in applying in clinical.

The authors have clarified some of their other data such as statistical significance in their in vivo studies which is helpful.

Reply: Thank you very much for your valuable and professional comments on our work. Thanks so much for the positive evaluation.